# Continuous Latent Process Flows

**Ruizhi Deng**[1,2*]  **Marcus A. Brubaker**[1,3,4]  **Greg Mori**[1,2]  **Andreas M. Lehrmann**[1]
[1]Borealis AI   [2]Simon Fraser University   [3]York University   [4]Vector Institute

## Abstract

Partial observations of continuous time-series dynamics at arbitrary time stamps exist in many disciplines. Fitting this type of data using statistical models with continuous dynamics is not only promising at an intuitive level but also has practical benefits, including the ability to generate continuous trajectories and to perform inference on previously unseen time stamps. Despite exciting progress in this area, the existing models still face challenges in terms of their representation power and the quality of their variational approximations. We tackle these challenges with continuous latent process flows (CLPF), a principled architecture decoding continuous latent processes into continuous observable processes using a time-dependent normalizing flow driven by a stochastic differential equation. To optimize our model using maximum likelihood, we propose a novel piecewise construction of a variational posterior process and derive the corresponding variational lower bound using importance weighting of trajectories. An ablation study demonstrates the effectiveness of our contributions and comparisons to state-of-the-art baselines show our model's favourable performance on both synthetic and real-world data.

## 1 Introduction

Sparse and irregular observations of continuous dynamics are common in many areas of science, including finance [15, 36], healthcare [16], and physics [30]. Time-series models driven by stochastic differential equations (SDEs) provide an elegant framework for this challenging scenario and have recently gained popularity in the machine learning community [11, 18, 24]. The SDEs are typically implemented by neural networks with trainable parameters and the latent processes defined by the SDEs are then decoded into an observable space with complex structure. Due to the lack of closed-form transition densities for most SDEs, dedicated variational approximations have been developed to maximize the observational log-likelihoods [2, 18, 24].

Despite the progress of existing works, challenges still remain for SDE-based models to be applied to irregular time-series data. One major challenge is the model's representation power. The continuous-time flow process (CTFP; [11]) uses a series of invertible mappings continuously indexed by time to transform a simple Wiener process to a more complex stochastic process. The use of a simple latent process and invertible transformations permits CTFP models to evaluate the exact likelihood of observations on any time grid efficiently, but they also limit the set of stochastic processes that CTFP can express to some specific form which can be obtained using Ito's Lemma and excludes many commonly seen stochastic processes. Another constraint of representation power in practice is the Lipschitz property of transformations in the models. The latent SDE model proposed by Hasan et al. [18] and CTFP both transform a latent stochastic process with constant variance to an observable one using injective mappings. Due to the Lipschitz property existing in many invertible neural network architectures, some processes that can be written as a non-Lipschitz transformation of a simple process, like geometric Brownian motion, cannot be expressed by these models unless specific choices of non-Lipschitz decoders are used. Apart from the model's representation power, variational inference is another challenge in training SDE-based models. The latent SDE model in the work of Li et al. [24]

---

[*]This work was done during an internship at Borealis AI. Correspondance to wsdmdeng@gmail.com.

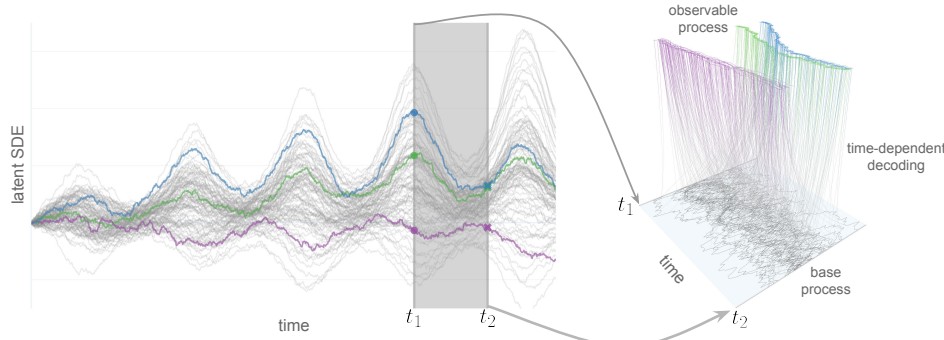

Figure 1: **Overview.** Our architecture uses a stochastic differential equation (SDE; left) to drive a time-dependent normalizing flow (NF; right). At time $t1$, $t2$ (grey bars), the values of the SDE trajectories (colored trajectories on the left) serve as conditioning information for the decoding of a simple base process (grey trajectories on the right) into a complex observable process (colored trajectories on the right). The color gradient on the right shows the individual trajectories of this transformation, which is driven by an augmented neural ODE. Since all stochastic processes and mappings are time-continuous, we can model observed data as partial realizations of a continuous process, enabling modelling of continuous dynamics and inference on irregular time grids.

uses a principled method of variational approximation based on importance weighting of trajectories between a variational posterior and a prior process. The variational posterior process is constructed using a single SDE conditioned on all observations and is therefore restricted to be a Markov process. This approach may lack the flexibility to approximate the true posterior process well enough in complex inference tasks, e.g., in an online setting or with variable-length observation sequences.

In this work we propose Continuous Latent Process Flows (CLPFs)[2], a model that is governed by latent dynamics defined by an expressive generic stochastic differential equation. Inspired by [11], we then use dynamic normalizing flows to decode each latent trajectory into a continuous observable process. Driven by different trajectories of the latent stochastic process continuously evolving with time, the dynamic normalizing flow can map a simple base process to a diverse class of observable processes. We illustrate this process in Fig. 1. This decoding is critical for the model to generate continuous trajectories and be trained to fit observations on irregular time grids using a variational approximation. Good variational approximations and proper handling of complex inference tasks like online inference depend on a flexible variational posterior process. Therefore, we also propose a principled method of defining and sampling from a non-Markov variational posterior process that is based on a piecewise evaluation of SDEs and can adapt to new observations. The proposed model excels at fitting observations on irregular time grids, generalizing to observations on more dense time grids, and generating trajectories continuous in time.

**Contributions.** In summary, we make the following contributions: (1) We propose a flow-based decoding of a generic SDE as a principled framework for continuous dynamics modeling of irregular time-series data. (2) We improve the variational approximation of the observational likelihood through a flexible non-Markov posterior process based on a piecewise evaluation of the underlying SDE; (3) We validate the effectiveness of our contributions in a series of ablation studies and comparisons to state-of-the-art time-series models, both on synthetic and real-world datasets.

## 2 Preliminaries

### 2.1 Stochastic Differential Equations

SDEs can be viewed as a stochastic analogue of ordinary differential equations (ODEs) in the sense that $\frac{d\boldsymbol{Z}_t}{dt} = \boldsymbol{\mu}(\boldsymbol{Z}_t, t) + \text{random noise} \cdot \boldsymbol{\sigma}(\boldsymbol{Z}_t, t)$. Let $\boldsymbol{Z}_t$ be a variable which continuously evolves with time. An $m$-dimensional SDE describing the stochastic dynamics of $\boldsymbol{Z}_t$ usually takes the form

$$\mathrm{d}\boldsymbol{Z}_t = \boldsymbol{\mu}(\boldsymbol{Z}_t, t)\,\mathrm{d}t + \boldsymbol{\sigma}(\boldsymbol{Z}_t, t)\,\mathrm{d}\boldsymbol{W}_t, \tag{1}$$

where $\boldsymbol{\mu}$ maps to an $m$-dimensional vector, $\boldsymbol{\sigma}$ is an $m \times k$ matrix, and $\boldsymbol{W}_t$ is a $k$-dimensional Wiener process. The solution of an SDE is a continuous-time stochastic process $\boldsymbol{Z}_t$ that satisfies the integral equation $\boldsymbol{Z}_t = \boldsymbol{Z}_0 + \int_0^t \boldsymbol{\mu}(\boldsymbol{Z}_s, s)\,\mathrm{d}s + \int_0^t \boldsymbol{\sigma}(\boldsymbol{Z}_s, s)\,\mathrm{d}\boldsymbol{W}_s$ with initial condition $\boldsymbol{Z}_0$, where

---

[2]Code available at `https://github.com/BorealisAI/continuous-latent-process-flows`

the stochastic integral should be interpreted as a traditional Itô integral [27, Chapter 3.1]. For each sample trajectory $\boldsymbol{\omega} \sim \boldsymbol{W}_t$, the stochastic process $\boldsymbol{Z}_t$ maps $\boldsymbol{\omega}$ to a different trajectory $\boldsymbol{Z}_t(\boldsymbol{\omega})$.

**Latent Dynamics and Variational Bound.** SDEs have been used as models of latent dynamics in a variety of contexts [2, 18, 24]. As closed-form finite-dimensional solutions to SDEs are rare, variational approximations are often used in practice. Li et al. [24] propose a principled way of re-weighting latent SDE trajectories for variational approximations using Girsanov's theorem [27, Chapter 8.6]. Specifically, consider a prior process and a variational posterior process in the interval $[0, T]$ defined by two stochastic differential equations $\mathrm{d}\boldsymbol{Z}_t = \boldsymbol{\mu_1}(\boldsymbol{Z}_t, t) \, \mathrm{d}t + \boldsymbol{\sigma}(\boldsymbol{Z}_t, t) \, \mathrm{d}\boldsymbol{W}_t$ and $\mathrm{d}\hat{\boldsymbol{Z}}_t = \boldsymbol{\mu_2}(\hat{\boldsymbol{Z}}_t, t) \, \mathrm{d}t + \boldsymbol{\sigma}(\hat{\boldsymbol{Z}}_t, t) \, \mathrm{d}\boldsymbol{W}_t$, respectively. Furthermore, let $p(\boldsymbol{x}|\boldsymbol{Z}_t)$ denote the probability of observing $\boldsymbol{x}$ conditioned on the trajectory of the latent process $\boldsymbol{Z}_t$ in the interval $[0, T]$. If there exists a mapping $\boldsymbol{u} : \mathbb{R}^m \times [0, T] \to \mathbb{R}^k$ such that $\boldsymbol{\sigma}(\boldsymbol{z}, t)\boldsymbol{u}(\boldsymbol{z}, t) = \boldsymbol{\mu_2}(\boldsymbol{z}, t) - \boldsymbol{\mu_1}(\boldsymbol{z}, t)$ and $\boldsymbol{u}$ satisfies Novikov's condition [27, Chapter 8.6], we obtain the variational lower bound

$$\log p(\boldsymbol{x}) = \log \mathbb{E}[p(\boldsymbol{x}|\boldsymbol{Z}_t)] = \log \mathbb{E}[p(\boldsymbol{x}|\hat{\boldsymbol{Z}}_t)\boldsymbol{M}_T] \geq \mathbb{E}[\log p(\boldsymbol{x}|\hat{\boldsymbol{Z}}_t) + \log \boldsymbol{M}_T], \qquad (2)$$

where $\boldsymbol{M}_T = \exp(-\int_0^T \frac{1}{2} \left| \boldsymbol{u}(\hat{\boldsymbol{Z}}_t, t) \right|^2 \mathrm{d}t - \int_0^T \boldsymbol{u}(\hat{\boldsymbol{Z}}_t, t)^T \, \mathrm{d}\boldsymbol{W}_t)$. See [24] for a formal proof.

## 2.2 Normalizing Flows

Normalizing flows [3, 8, 12, 13, 21, 22, 23, 28, 29, 31] employ a bijective mapping $f : \mathbb{R}^d \to \mathbb{R}^d$ to transform a random variable $\boldsymbol{Y}$ with a simple base distribution $p_{\boldsymbol{Y}}$ to a random variable $\boldsymbol{X}$ with a complex target distribution $p_{\boldsymbol{X}}$. We can sample from a normalizing flow by first sampling $\boldsymbol{y} \sim p_{\boldsymbol{Y}}$ and then transforming it to $\boldsymbol{x} = f(\boldsymbol{y})$. As a result of invertibility, normalizing flows can also be used for density estimation. Using the change-of-variables formula, we have $\log p_{\boldsymbol{X}}(\boldsymbol{x}) = \log p_{\boldsymbol{Y}}(g(\boldsymbol{x})) + \log \left| \det \left( \frac{\partial g}{\partial \boldsymbol{x}} \right) \right|$, where $g$ is the inverse of $f$.

**Continuous Indexing.** More recently, normalizing flows have been augmented with a continuous index [6, 10, 11]. For instance, the continuous-time flow process (CTFP; [11]) models irregular observations of a continuous-time stochastic process. Specifically, CTFP transforms a simple $d$-dimensional Wiener process $\boldsymbol{W}_t$ to another continuous stochastic process $\boldsymbol{X}_t$ using the transformation $\boldsymbol{X}_t = f(\boldsymbol{W}_t, t)$, where $f(\boldsymbol{w}, t)$ is an invertible mapping for each $t$. Despite its benefits of exact log-likelihood computation of arbitrary finite-dimensional distributions, the expressive power of CTFP to model stochastic processes is limited in the following two aspects: (1) An application of Itô's lemma [27, Chapter 4.2] shows that CTFP can only represent stochastic processes of the form

$$\mathrm{d}f(\boldsymbol{W}_t, t) = \{\frac{\mathrm{d}f}{\mathrm{d}t}(\boldsymbol{W}_t, t) + \frac{1}{2} \mathrm{Tr}(\mathbf{H}_{\boldsymbol{w}}f(\boldsymbol{W}_t, t))\} \, \mathrm{d}t + (\nabla_{\boldsymbol{w}}f^T(\boldsymbol{W}_t, t))^T \, \mathrm{d}\boldsymbol{W}_t, \qquad (3)$$

where $\mathbf{H}_{\boldsymbol{w}}f$ is the Hessian matrix of $f$ with respect to $\boldsymbol{w}$ and $\nabla_{\boldsymbol{w}}f$ is the derivative. A variety of stochastic processes, from simple processes like the Ornstein-Uhlenbeck (OU) process to more complex non-Markov processes, fall outside of this limited class and cannot be learned using CTFP (see Appendix A for formal proofs); (2) Many normalizing flow architectures are compositions of Lipschitz-continuous transformations [7, 8, 17]. It is therefore challenging to model certain stochastic processes that are non-Lipschitz transformations of simple processes using CTFP without prior knowledge about the functional form of the observable processes and custom-tailored normalizing flows with non-Lipschitz transformations (see Appendix B for an example).

A latent variant of CTFP is further augmented with a static latent variable to introduce non-Markovian behavior. It models continuous stochastic processes as $\boldsymbol{X}_t = f(\boldsymbol{W}_t, t; \boldsymbol{Z})$, where $\boldsymbol{Z}$ is a latent variable with standard Gaussian distribution and $f(\cdot, \cdot; z)$ is a CTFP model that decodes each sample $z$ of $\boldsymbol{Z}$ into a stochastic process with continuous trajectories. Latent CTFP can be used to estimate finite-dimensional distributions using a variational approximation. However, it is unclear how much a latent variable $\boldsymbol{Z}$ with finite dimensions can improve CTFPs representation power.

## 3 Model

Equipped with these tools, we can now describe our problem setting and model architecture. Let $\{(\boldsymbol{x}_{t_i}, t_i)\}_{i=1}^n$ denote a sequence of $d$-dimensional observations sampled at arbitrary points in time,

where $\boldsymbol{x}_{t_i}$ and $t_i$ denote the value and time stamp of the observation, respectively. The observations are assumed to be partial realizations of a continuous-time stochastic process $\boldsymbol{X}_t$. Our training objective is the maximization of the observational log-likelihood induced by $\boldsymbol{X}_t$ on a given time grid,

$$\mathcal{L} = \log p_{\boldsymbol{X}_{t_1}, \ldots, \boldsymbol{X}_{t_n}}(\boldsymbol{x}_{t_1}, \ldots, \boldsymbol{x}_{t_n}), \tag{4}$$

for an inhomogeneous collection of sequences with varying lengths and time stamps. At test-time, in addition to the maximization of log-likelihoods, we are also interested in sampling sparse, dense, or irregular trajectories with finite-dimensional distributions that conform with these log-likelihoods. We model this challenging scenario with Continuous Latent Process Flows (CLPF). In Section 3.1, we present our model in more detail. Training and inference methods will be discussed in Section 3.2.

## 3.1 Continuous Latent Process Flows

A Continuous Latent Process Flow consists of two major components: an SDE describing the continuous latent dynamics of an observable stochastic process and a continuously indexed normalizing flow serving as a time-dependent decoder. The architecture of the normalizing flow itself can be specified in multiple ways, e.g., as an augmented neural ODE [14] or as a series of affine transformations [10]. The following paragraphs discuss the relationship between these components in more detail.

**Continuous Latent Dynamics.** Analogous to our overview in Section 2.1, we model the evolution of an $m$-dimensional time-continuous latent state $\boldsymbol{Z}_t$ in the time interval $[0, T]$ using a flexible stochastic differential equation driven by an $m$-dimensional Wiener Process $\boldsymbol{W}_t$,

$$\mathrm{d}\boldsymbol{Z}_t = \boldsymbol{\mu}_\gamma(\boldsymbol{Z}_t, t) \, \mathrm{d}t + \boldsymbol{\sigma}_\gamma(\boldsymbol{Z}_t, t) \, \mathrm{d}\boldsymbol{W}_t, \tag{5}$$

where $\gamma$ denotes the (shared) learnable parameters of the drift function $\boldsymbol{\mu}$ and variance function $\boldsymbol{\sigma}$. In our experiments, we implement $\boldsymbol{\mu}$ and $\boldsymbol{\sigma}$ using deep neural networks (see Appendix E for details). Importantly, the latent state $\boldsymbol{Z}_t$ exists for each $t \in [0, T]$ and can be sampled on any given time grid, which can be irregular and different for each sequence.

**Time-Dependent Decoding.** Latent variable models decode a latent state into an observable variable with complex distribution. As an observed sequence $\{(\boldsymbol{x}_{t_i}, t_i)\}_{i=1}^n$ is assumed to be a partial realization of a continuous-time stochastic process, continuous trajectories of the latent process $\boldsymbol{Z}_t$ should be decoded into continuous trajectories of the observable process $\boldsymbol{X}_t$, and not discrete distributions. Following recent advances in dynamic normalizing flows [6, 10, 11], we model $\boldsymbol{X}_t$ as

$$\boldsymbol{X}_t = F_\theta(\boldsymbol{O}_t; \boldsymbol{Z}_t, t), \tag{6}$$

where $\boldsymbol{O}_t$ is a $d$-dimensional stochastic process with closed-form transition density[3] and $F_\theta(\,\cdot\,; \boldsymbol{z}_t, t)$ is a normalizing flow parameterized by $\theta$ for any $\boldsymbol{z}_t, t$. The transformation $F_\theta$ decodes each sample path of $\boldsymbol{Z}_t$ into a complex distribution over continuous trajectories $\boldsymbol{X}_t$ if $F_\theta$ is a continuous mapping and the sampled trajectories of the base process $\boldsymbol{O}_t$ are continuous with respect to time $t$. Unlike [11], who use a Wiener process as base process, we use the Ornstein–Uhlenbeck (OU) process, which has a stationary marginal distribution and bounded variance. As a result, the variance of the observation process does not increase due to the increase of variance in the base process and is primarily determined by the latent process $\boldsymbol{Z}_t$ and flow transformation $F_\theta$.

**Flow Architecture.** The continuously indexed normalizing flow $F_\theta(\,\cdot\,; \boldsymbol{z}_t, t)$ can be implemented in multiple ways. Deng et al. [11] use ANODE [14], defined as the solution to the initial value problem

$$\frac{\mathrm{d}}{\mathrm{d}\tau} \begin{pmatrix} \boldsymbol{h}(\tau) \\ \boldsymbol{a}(\tau) \end{pmatrix} = \begin{pmatrix} f_\theta(\boldsymbol{h}(\tau), \boldsymbol{a}(\tau), \tau) \\ g_\theta(\boldsymbol{a}(\tau), \tau) \end{pmatrix}, \quad \begin{pmatrix} \boldsymbol{h}(\tau_0) \\ \boldsymbol{a}(\tau_0) \end{pmatrix} = \begin{pmatrix} \boldsymbol{o}_t \\ (\boldsymbol{z}_t, t)^T \end{pmatrix}, \tag{7}$$

where $\tau \in [\tau_0, \tau_1]$, $\boldsymbol{h}(\tau) \in \mathbb{R}^d$, $\boldsymbol{a}(\tau) \in \mathbb{R}^{m+1}$, $f_\theta : \mathbb{R}^d \times \mathbb{R}^{m+1} \times [\tau_0, \tau_1] \to \mathbb{R}^d$, $g_\theta : \mathbb{R}^{m+1} \times [t_0, t_1] \to \mathbb{R}$, and $F_\theta$ is defined as the solution of $\boldsymbol{h}(\tau)$ at $\tau = \tau_1$. Note the difference between $t$ and $\tau$: while $t \in [0, T]$ describes the continuous process dynamics, $\tau \in [\tau_0, \tau_1]$ describes the continuous time-dependent decoding at each time step $t$. Alternatively, Cornish et al. [10] propose a variant of continuously indexed normalizing flows based on a series of $N$ affine transformations $f_i$,

$$\boldsymbol{h}_t^{(i+1)} = f_i(\boldsymbol{h}_t^{(i)}; \boldsymbol{z}_t, t) = k(\boldsymbol{h}_t^{(i)} \cdot \exp(-u^{(i)}(\boldsymbol{z}_t, t)) - v^{(i)}(\boldsymbol{z}_t, t)), \tag{8}$$

---

[3]More precisely, the conditional distribution $p_{\boldsymbol{O}_{t_i} | \boldsymbol{O}_{t_j}}(\boldsymbol{o}_{t_i} | \boldsymbol{o}_{t_j})$ must exist in closed form for any $t_j < t_i$.

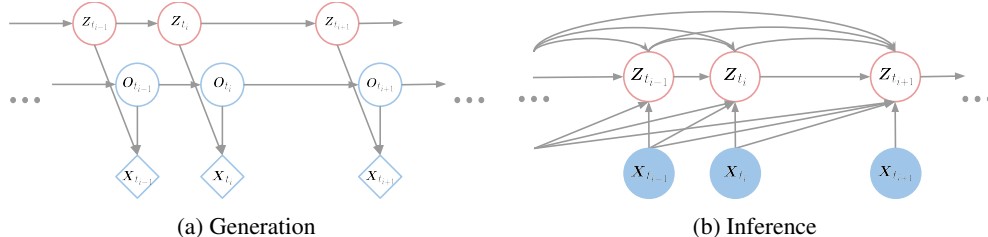

|  (a) Generation | (b) Inference |

Figure 2: **Graphical Model of Generation and Inference.** Red circles represent latent variables $\boldsymbol{Z}_{t_i}$. Unfilled blue circles represent samples from the OU process at discrete time points $\boldsymbol{O}_{t_i}$. Blue diamonds in Fig. 2a indicate each $\boldsymbol{X}_{t_i}$ is the result of a deterministic mapping of $\boldsymbol{Z}_{t_i}$ and $\boldsymbol{O}_{t_i}$. Filled blue circles in Fig. 2b represent observed variables $\boldsymbol{X}_{t_i}$.

where $\boldsymbol{h}_t^{(0)} = \boldsymbol{o}_t$, $\boldsymbol{h}_t^{(N)} = \boldsymbol{x}_t$, $u^{(i)}$ and $v^{(i)}$ are unconstrained transformations, and $k$ is an invertible mapping like a residual flow [8]. The temporal relationships among $\boldsymbol{Z}_t$, $\boldsymbol{O}_t$, and $\boldsymbol{X}_t$ from a graphical model point of view are shown in Fig. 2a.

## 3.2 Training and Inference

With the model fully specified, we can now focus our attention on training and inference. Computing the observational log-likelihood (Eq.(4)) induced by a time-dependent decoding of an SDE (Eq.(6)) on an arbitrary time grid is challenging, because only few SDEs have closed-form transition densities. Consequently, variational approximations are needed for flexible SDEs such as Eq.(5). We propose a principled way of approximating the observational log-likelihood with a variational lower bound based on a novel piecewise construction of the posterior distribution of the latent process. In summary, we first express the observational log-likelihood as a marginalization over piecewise factors conditioned on a latent trajectory, then approximate this intractable computation with a piecewise variational posterior process, and finally derive a variational lower bound for it.

**Observational Log-Likelihood.** The observational log-likelihood can be written as an expectation over latent trajectories of the conditional likelihood, which can be evaluated in closed form,

$$
\begin{aligned}
\mathcal{L} &= \log p_{\boldsymbol{X}_{t_1},\ldots,\boldsymbol{X}_{t_n}}(\boldsymbol{x}_{t_1},\ldots,\boldsymbol{x}_{t_n}) \\
&= \log \mathbb{E}_{\omega \sim \boldsymbol{W}_t}\left[ p_{\boldsymbol{X}_{t_1},\ldots,\boldsymbol{X}_{t_n}|\boldsymbol{Z}_t}(\boldsymbol{x}_{t_1},\ldots,\boldsymbol{x}_{t_n}|\boldsymbol{Z}_t(\omega)) \right] \\
&= \log \mathbb{E}_{\omega \sim \boldsymbol{W}_t}\left[ \prod_{i=1}^n p_{\boldsymbol{X}_{t_i}|\boldsymbol{X}_{t_{i-1}},\boldsymbol{Z}_{t_i},\boldsymbol{Z}_{t_{i-1}}}(\boldsymbol{x}_{t_i}|\boldsymbol{x}_{t_{i-1}},\boldsymbol{Z}_{t_i}(\omega),\boldsymbol{Z}_{t_{i-1}}(\omega)) \right],
\end{aligned}
\tag{9}
$$

where $\boldsymbol{Z}_t(\omega)$ denotes a sample trajectory of $\boldsymbol{Z}_t$ driven by $\omega \sim \boldsymbol{W}_t$. For simplicity, we assume w.l.o.g. and in this section only $\boldsymbol{Z}_0, \boldsymbol{X}_0$ to be given. As a result of invertibility, the conditional likelihood terms $p_{\boldsymbol{X}_{t_i}|\boldsymbol{X}_{t_{i-1}},\boldsymbol{Z}_{t_i},\boldsymbol{Z}_{t_{i-1}}}$ can be computed using the change-of-variables formula,

$$
\begin{aligned}
&\log p_{\boldsymbol{X}_{t_i}|\boldsymbol{X}_{t_{i-1}},\boldsymbol{Z}_{t_i},\boldsymbol{Z}_{t_{i-1}}}(\boldsymbol{x}_{t_i}|\boldsymbol{x}_{t_{i-1}},\boldsymbol{Z}_{t_i}(\omega),\boldsymbol{Z}_{t_{i-1}}(\omega)) \\
&= \log p_{\boldsymbol{O}_{t_i}|\boldsymbol{O}_{t_{i-1}}}(\boldsymbol{o}_{t_i}|\boldsymbol{o}_{t_{i-1}}) - \log \left| \det \frac{\partial F_\theta(\boldsymbol{o}_{t_i};\boldsymbol{Z}_{t_i}(\omega),t_i)}{\partial \boldsymbol{o}_{t_i}} \right|,
\end{aligned}
\tag{10}
$$

where $\boldsymbol{o}_{t_i} = F_\theta^{-1}(\boldsymbol{x}_{t_i};\boldsymbol{Z}_{t_i}(\omega),t_i)$.

**Piecewise Construction of Variational Posterior.** Directly computing the observational log-likelihood is intractable and we use a variational approximation during both training and inference. Good variational approximations rely on variational posteriors that are close enough to the true posterior of the latent trajectory conditioned on observations. Previous methods [24] use a single SDE to propose the variational posterior conditioned on all observations. Instead, we develop a more flexible method that can update the posterior process parameters when a new observation is seen and naturally adapts to different time grids. Our posterior process is not constrained by the Markov property of SDE solutions. Moreover, the proposed method serves as the basis for a principled approach to online inference tasks using variational posterior processes.

Our construction makes use of a further decomposition of the observational log-likelihood based on the following facts: $\{\boldsymbol{W}_{s+t} - \boldsymbol{W}_s\}_{t \geqslant 0}$ is also a Wiener process $\forall s \geqslant 0$ and the solution to Eq.(5) is a Markov process. Specifically, let $\{(\boldsymbol{\Omega}^{(i)}, \boldsymbol{\mathcal{F}}_{t_i-t_{i-1}}^{(i)}, P^{(i)})\}_{i=1}^n$ be a series of probability spaces on which $n$ independent $m$-dimensional Wiener processes $\boldsymbol{W}_t^{(i)}$ are defined. We can sample a complete trajectory of the Wiener process $\boldsymbol{W}_t$ in the interval $[0, T]$ by sampling independent trajectories of length $t_i - t_{i-1}$ from $\boldsymbol{\Omega}^{(i)}$ and adding them, i.e., $\omega_t = \sum_{\{i:t_i<t\}} \omega_{t_i-t_{i-1}}^{(i)} + \omega_{t-t_{i^*-1}}^{(i^*)}$, where $i^* = \max\{i : t_i < t\} + 1$. As a result, we can solve the latent stochastic differential equations in a piecewise manner. Samples of $\boldsymbol{Z}_{t_i}$ can be obtained by solving the stochastic differential equation

$$\mathrm{d}\widehat{\boldsymbol{Z}}_t = \boldsymbol{\mu}_\gamma(\widehat{\boldsymbol{Z}}_t, t + t_{i-1})\,\mathrm{d}t + \boldsymbol{\sigma}_\gamma(\widehat{\boldsymbol{Z}}_t, t + t_{i-1})\,\mathrm{d}\boldsymbol{W}_t^{(i)} \tag{11}$$

in the interval $[0, t_i - t_{i-1}]$, with $\boldsymbol{Z}_{t_{i-1}}$ as the initial condition. The log-likelihood of the observations can thus be rewritten as

$$\begin{aligned}
\mathcal{L} &= \log \mathbb{E}_{\omega^{(1)},\ldots,\omega^{(n)} \sim \boldsymbol{W}_t^{(1)} \times \cdots \times \boldsymbol{W}_t^{(n)}} \left[ \prod_{i=1}^n p(\boldsymbol{x}_{t_i} | \boldsymbol{x}_{t_{i-1}}, \boldsymbol{z}_{t_i}, \boldsymbol{z}_{t_{i-1}}, \omega^{(i)}) \right] \\
&= \log \mathbb{E}_{\omega^{(1)} \sim \boldsymbol{W}_t^{(1)}} \left[ p(\boldsymbol{x}_{t_1} | \boldsymbol{x}_{t_0}, \boldsymbol{z}_{t_1}, \boldsymbol{z}_{t_0}, \omega^{(1)}) \ldots \right. \\
&\qquad\qquad \left. \mathbb{E}_{\omega^{(i)} \sim \boldsymbol{W}_t^{(i)}} \left[ p(\boldsymbol{x}_{t_i} | \boldsymbol{x}_{t_{i-1}}, \boldsymbol{z}_{t_i}, \boldsymbol{z}_{t_{i-1}}, \omega^{(i)}) \mathbb{E}_{\omega^{(i+1)} \sim \boldsymbol{W}_t^{(i+1)}} [\ldots] \ldots \right] \right].
\end{aligned} \tag{12}$$

It is worth noting that the value of $\boldsymbol{x}_{t_i}$ is determined by $\boldsymbol{z}_{t_i}$ and $\boldsymbol{o}_{t_i}$. With the distribution of $\boldsymbol{O}_{t_i}$ depending on the value of $\boldsymbol{o}_{t_{i-1}}$, and thus on $\boldsymbol{z}_{t_{i-1}}$ and $\boldsymbol{x}_{t_{i-1}}$, after marginalizing over $\boldsymbol{o}_{t_i}$ each conditional likelihood term in Eq.(12) is conditioned on $\boldsymbol{x}_{t_{i-1}}$, $\boldsymbol{z}_{t_i}$, and $\boldsymbol{z}_{t_{i-1}}$. For simplicity we also omit conditioning of each term on Wiener process trajectories up to $t_{i-1}$, i.e., $\omega^{(1)}, \omega^{(2)}, \ldots, \omega^{(i)}$. In preparation of our variational approximation we can now introduce one posterior SDE

$$\mathrm{d}\tilde{\boldsymbol{Z}}_t = \boldsymbol{\mu}_{\phi_i}(\tilde{\boldsymbol{Z}}_t, t + t_{i-1})\,\mathrm{d}t + \boldsymbol{\sigma}_\gamma(\tilde{\boldsymbol{Z}}_t, t + t_{i-1})\,\mathrm{d}\boldsymbol{W}_t^{(i)} \tag{13}$$

for each expectation $\mathbb{E}_{\omega^{(i)} \sim \boldsymbol{W}_t^{(i)}} \left[ p(\boldsymbol{x}_{t_i} | \boldsymbol{x}_{t_{i-1}}, \boldsymbol{z}_{t_i}, \boldsymbol{z}_{t_{i-1}}, \omega^{(i)}) \mathbb{E}_{\omega^{(i+1)} \sim \boldsymbol{W}_t^{(i+1)}} [\ldots] \ldots \right]$ in Eq.(12).

**Variational Lower Bound with Piecewise Importance Weighting.** The posterior SDEs in Eq.(13) form the basis for a variational approximation of the expectations in Eq.(12). Specifically, sampling $\tilde{z}$ from the posterior SDE, the expectation can be rewritten as

$$\mathbb{E}_{\omega^{(i)} \sim \boldsymbol{W}_t^{(i)}} \left[ p(\boldsymbol{x}_{t_i} | \boldsymbol{x}_{t_{i-1}}, \tilde{\boldsymbol{z}}_{t_i}, \boldsymbol{z}_{t_{i-1}}, \omega^{(i)}) \boldsymbol{M}^{(i)}(\omega^{(i)}) \mathbb{E}_{\omega^{(i+1)} \sim \boldsymbol{W}_t^{(i+1)}} [\ldots] \ldots \right], \tag{14}$$

where $\boldsymbol{M}^{(i)} = \exp(-\int_0^{t_i-t_{i-1}} \frac{1}{2} |\boldsymbol{u}(\tilde{\boldsymbol{Z}}_s, s)|^2 \,\mathrm{d}s - \int_0^{t_i-t_{i-1}} \boldsymbol{u}(\tilde{\boldsymbol{Z}}_s, s)^T \,\mathrm{d}\boldsymbol{W}_s^{(i)})$ serves as importance weight for the sampled trajectory between the prior latent SDE (Eq.(11)) and posterior latent SDE (Eq.(13)), with $\boldsymbol{u}$ satisfying $\boldsymbol{\sigma}_\gamma(\boldsymbol{z}, s + t_{i-1})\boldsymbol{u}(\boldsymbol{z}, s) = \boldsymbol{\mu}_{\phi_i}(\boldsymbol{z}, s + t_{i-1}) - \boldsymbol{\mu}_\gamma(\boldsymbol{z}, s + t_{i-1})$. We can define such a posterior latent SDE for each interval. As a result, a sample $\tilde{z}_{t_i}$ can be obtained by solving the posterior SDEs defined on the intervals up to $t_i$ given the initial value $\tilde{z}_{t_0}$ and sample paths of Wiener processes up to $t_i$, i.e., $\omega^{(1)}, \omega^{(2)}, \ldots, \omega^{(i)}$. After applying the importance weight to each interval and Jensen's inequality, we can derive the following evidence lower bound (ELBO) of the log-likelihood:

$$\begin{aligned}
\mathcal{L} &= \log \mathbb{E}_{\omega^{(1)} \sim \boldsymbol{W}_t^{(1)}} \left[ p(\boldsymbol{x}_{t_1} | \boldsymbol{x}_{t_0}, \tilde{\boldsymbol{z}}_{t_1}, \tilde{\boldsymbol{z}}_{t_0}, \omega^{(1)}) \boldsymbol{M}^{(1)} \ldots \mathbb{E}_{\omega^{(i)} \sim \boldsymbol{W}_t^{(i)}} \left[ p(\boldsymbol{x}_{t_i} | \boldsymbol{x}_{t_{i-1}}, \tilde{\boldsymbol{z}}_{t_i}, \tilde{\boldsymbol{z}}_{t_{i-1}}, \omega^{(i)}) \boldsymbol{M}^{(i)} \ldots \right] \right] \\
&= \log \mathbb{E}_{\omega^{(1)},\ldots,\omega^{(n)} \sim \boldsymbol{W}_t^{(1)} \times \cdots \times \boldsymbol{W}_t^{(n)}} \left[ \prod_{i=1}^n p(\boldsymbol{x}_{t_i} | \boldsymbol{x}_{t_{i-1}}, \tilde{\boldsymbol{z}}_{t_i}, \tilde{\boldsymbol{z}}_{t_{i-1}}, \omega^{(i)}) \boldsymbol{M}^{(i)}(\omega^{(i)}) \right] \\
&\geqslant \mathbb{E}_{\omega^{(1)},\ldots,\omega^{(n)} \sim \boldsymbol{W}_t^{(1)} \times \cdots \times \boldsymbol{W}_t^{(n)}} \left[ \sum_{i=1}^n \log p(\boldsymbol{x}_{t_i} | \boldsymbol{x}_{t_{i-1}}, \tilde{\boldsymbol{z}}_{t_i}, \tilde{\boldsymbol{z}}_{t_{i-1}}, \omega^{(i)}) + \sum_{i=1}^n \log \boldsymbol{M}^{(i)}(\omega^{(i)}) \right].
\end{aligned} \tag{15}$$

The bound above can be further extended into a tighter bound in IWAE [5] form by drawing multiple independent samples from each $\boldsymbol{W}_t^{(i)}$. The variational parameter $\phi_i$ is the output of an encoder RNN that takes as inputs the sequence of observations up to $t_i$, $\{\boldsymbol{x}_{t_1}, \ldots, \boldsymbol{x}_{t_i}\}$, and the sequence

of previously sampled latent states, $\{z_{t_1}, \ldots, z_{t_{i-1}}\}$. The parameter $\phi_i$ is thus updated based on both the latest observation and past history. As a result, the variational posterior distributions of the latent states $Z_{t_i}$ are no longer constrained to be Markov and the parameterization of the variational posterior can flexibly adapt to different time grids. The dependency structure between latent and observed variables during inference is depicted in Fig. 2b.

## 4   Related Work

The introduction of neural ODEs [7] unified modeling approaches based on differential equations and modern machine learning. Explorations of time-series modeling leveraging this class of approach have been conducted, which we review here.

The latent ODE model [7, 32] propagates a latent state across time using ordinary differential equations. As a result, the entire latent trajectory is solely determined by its initial value. Even though latent ODE models have continuous latent trajectories, the latent state is decoded into observations at each time step independently. Neural controlled differential equations (CDEs) [20] and rough differential equations (RDEs) [25] propagate a hidden state across time continuously using controlled differential equations driven by functions of time interpolated from observations on irregular time grids. Neural ODE Processes (NDPs) [26] construct a distribution over neural ODEs in order to effectively manage uncertainty over the underlying process that generates the data.

Among existing time-series works with continuous dynamics, the latent SDE model [24] is most similar to ours. The SDE model includes an adjoint sensitivity method for training SDEs; the derivation of the variational lower bound in our proposed model is based on the same principle of trajectory importance weighting between two stochastic differential equations. The posterior process there is defined as a global stochastic differential equation. Our model further exploits the given observation time grid of each sequence to induce a piecewise posterior process with richer structure.

The continuous-time flow process [11] (CTFP) models irregular time-series data as an incomplete realization of continuous-time stochastic processes obtained by applying normalizing flows to the Wiener process. We have discussed the limits of CTFP in Sec. 2.2. Because CTFP is a generative model that is guaranteed to generate continuous trajectories, we use it as the decoder of a latent process for better inductive bias in modeling continuous dynamics. Apart from CTFP, there are also works outside the deep learning literature that apply invertible transformations to stochastic processes [33, 34]. Warped Gaussian Processes [33] transform Gaussian processes to non-Gaussian processes in observation space using monotonic functions. Copula Processes [34] extend the concept of copulas from multivariate random variables to stochastic processes. They transform a stochastic process via a series of marginal cumulative distribution functions to obtain another process while preserving the underlying dependency structure.

Alternative training frameworks have also been explored for neural SDEs [18, 19]. Stochastic differential equations can be learned as latent dynamics with a variational approximation [18]. Connections between generative adversarial network (GAN) objectives and neural SDEs have been drawn [19]. Brownian motion inputs can be mapped to time-series outputs. Alongside a CDE-based discriminator GAN-based training can be conducted to obtain continuous-time generative models.

## 5   Experiments

We compare our proposed architecture against several baseline models with continuous dynamics that can be used to fit irregular time-series data, including CTFP, latent CTFP, latent SDE, and latent ODE. We also run experiments with Variational RNN (VRNN) [9], a model that can be used to fit sequential data. However, VRNN does not define a continuous dynamical model and cannot generate trajectories with finite-dimensional distributions that are consistent with its log-likelihood estimation. All implementation and training details can be found in the supplementary material.

### 5.1   Synthetic Data

We evaluate our model on synthetic data sampled from known stochastic processes to verify its ability to capture a variety of continuous dynamics. We use the following processes:

Table 1: **Quantitative Evaluation (Synthetic Data)**. We show test negative log-likelihoods (NLLs) of four synthetic stochastic processes across different models. Below each process, we indicate the intensity of the Poisson process from which the time stamps for the test sequences were sampled for testing. [GBM: geometric Brownian motion (ground truth NLLs: $[\lambda = 2, \lambda = 20] = [0.388, -0.788]$); LSDE: linear SDE; CAR: continuous auto-regressive process; SLC: stochastic Lorenz curve]

| Model | GBM | | LSDE | | CAR | | SLC | |
|---|---|---|---|---|---|---|---|---|
| | $\lambda = 2$ | $\lambda = 20$ | $\lambda = 2$ | $\lambda = 20$ | $\lambda = 2$ | $\lambda = 20$ | $\lambda = 20$ | $\lambda = 40$ |
| VRNN [9] | 0.425 | -0.650 | -0.634 | -1.665 | 1.832 | 2.675 | 2.237 | 1.753 |
| Latent ODE [32] | 1.916 | 1.796 | 0.900 | 0.847 | 4.872 | 4.765 | 9.117 | 9.115 |
| CTFP [11] | 2.940 | 0.678 | -0.471 | -1.778 | 383.593 | 51.950 | 0.489 | -0.586 |
| Latent CTFP [11] | 1.472 | -0.158 | -0.468 | -1.784 | 249.839 | 43.007 | 1.419 | -0.077 |
| Latent SDE [24] | 1.243 | 1.778 | 0.082 | 0.217 | 3.594 | 3.603 | 7.740 | 8.256 |
| CLPF (**ours**) | 0.444 | -0.698 | -0.831 | -1.939 | 1.322 | -0.077 | -2.620 | -3.963 |

**Geometric Brownian Motion** $\mathrm{d}\boldsymbol{X}_t = \mu\boldsymbol{X}_t \, \mathrm{d}t + \sigma\boldsymbol{X}_t \, \mathrm{d}\boldsymbol{W}_t$. Even though geometric Brownian motion can theoretically be captured by the CTFP model, it would require the normalizing flow to be non-Lipschitz; there is no such constraint for the proposed CLPF model.

**Linear SDE** $\mathrm{d}\boldsymbol{X}_t = (a(t)\boldsymbol{X}_t + b(t)) \, \mathrm{d}t + \sigma(t) \, \mathrm{d}\boldsymbol{W}_t$. The drift term of the SDE is a linear transformation of $\boldsymbol{X}_t$ and the variance term is a deterministic function of time $t$. An application of Itô's lemma shows that the solution is a stochastic process that cannot be captured by CTFP.

**Continuous AR(4) Process.** This process tests our model's ability to capture non-Markov processes (see Appendix C for implementation details):

$$\boldsymbol{X}_t = [d, 0, 0, 0]\boldsymbol{Y}_t, \\ \mathrm{d}\boldsymbol{Y}_t = A\boldsymbol{Y}_t \, \mathrm{d}t + e \, \mathrm{d}\boldsymbol{W}_t, \qquad \text{where } A = \begin{bmatrix} 0 & 1 & 0 & 0 \\ 0 & 0 & 1 & 0 \\ 0 & 0 & 0 & 1 \\ a_1 & a_2 & a_3 & a_4 \end{bmatrix}. \tag{16}$$

**Stochastic Lorenz Curve.** We evaluate our model's performance on multi-dimensional (3D) data:

$$\mathrm{d}\boldsymbol{X}_t = \sigma(\boldsymbol{Y}_t - \boldsymbol{X}_t) \, \mathrm{d}t + \alpha_x \, \mathrm{d}\boldsymbol{W}_t, \\ \mathrm{d}\boldsymbol{Y}_t = (\boldsymbol{X}_t(\rho - \boldsymbol{Z}_t) - \boldsymbol{Y}_t) \, \mathrm{d}t + \alpha_y \, \mathrm{d}\boldsymbol{W}_t, \\ \mathrm{d}\boldsymbol{Z}_t = (\boldsymbol{X}_t\boldsymbol{Y}_t - \beta\boldsymbol{Z}_t) \, \mathrm{d}t + \alpha_z \, \mathrm{d}\boldsymbol{W}_t. \tag{17}$$

In all cases, we sample the observation time stamps from a homogeneous Poisson process with intensity $\lambda$. To demonstrate our model's ability to generalize to different time grids, we evaluate it using different intensities $\lambda$. An approximate numerical solution to the SDEs is obtained using the Euler-Maruyama scheme for the Itô integral.

Our results on synthetic data are displayed in Table 1. We train and evaluate all models on observations of sample trajectories in the interval $[0, 30]$ with observation time stamps sampled from a Poisson process with intensity $\lambda = 2$, except for the stochastic Lorenz curve which is sampled in the interval $[0, 2]$. The results demonstrate the proposed CLPF model's favourable performance across the board. In particular, our model outperforms both CTFP variants on all four synthetic datasets. We attribute this competitive edge to the expressive power of a generic SDE over a static latent variable. The CTFP models also perform relatively poorly on the Continuous AR(4) process, which is non-Markov. As the Continuous AR(4) process has an underlying 4-dim. stochastic process $\boldsymbol{Y}_t$ and is generated by projecting $\boldsymbol{Y}_t$ to a 1-dim. observation space $\boldsymbol{X}_t$, models driven by a high-dimensional latent process like latent SDE and CLPF show better performance in this case.

To evaluate our model's performance in capturing continuous dynamics, we increase the (average) density of observations and generate observation time stamps from a Poisson process with intensity $\lambda = 20$ ($\lambda = 40$ for SLC). The results show that models that can generate continuous trajectories, including CTFP, latent CTFP and CLPF, generalize better to dense observations than the other models. In the first row of Fig. 3, we visualize sample sequences from CLPF and VRNN models trained on LSDE data and compare them with samples from an LSDE ground truth process. We run both CLPF and VRNN on a time grid between 0 and 30, with a gap of 0.01 between steps. Samples from the CLPF models share important visual properties with samples from the ground truth process. On the other hand, VRNN fails to generate sequences visually similar to the ground truth, despite its competitive density estimation results on sparse time grids.

Table 2: **Likelihood Estimation (Real-World Data).** We show test negative log-likelihoods (NLLs; lower is better). For CTFP, the reported values are exact; for the other models, we report IWAE bounds using $K = 125$ samples. CLPF-ANODE stands for a CLPF model implemented with augmented neural ODEs. CLPF-iRes stands for a CLPF model implemented with indexed residual flows.

| Model | Mujoco [32] | BAQD [35] | PTBDB [4] |
|---|---|---|---|
| VRNN [9] | -15.876 | -1.204 | -2.035 |
| Latent ODE [32] | 23.551 | 2.540 | -0.533 |
| Latent SDE [24] | 3.071 | 1.512 | -1.358 |
| CTFP [11] | -7.598 | -0.170 | -1.281 |
| Latent CTFP [11] | -12.693 | -0.480 | -1.659 |
| CLPF-ANODE (**ours**) | -14.694 | -0.619 | -1.575 |
| CLPF-iRes (**ours**) | -10.873 | -0.486 | -1.519 |

Table 3: **Ablation Study (Synthetic Data).** We show test negative log-likelihoods across different variants of the proposed model. We report IWAE bounds using $K = 125$ samples with observation time stamps sampled from a Poisson point process with $\lambda = 2$. [CLPF-Global: single global posterior SDE in latent SDE style [24]; CLPF-Independent: independent decoder instead of CTFP-decoder; CLPF-Wiener: Wiener base process instead of OU-process]

| Model | GBM | LSDE | CAR | SLC |
|---|---|---|---|---|
| CLPF-Global | 0.447 | -0.821 | 1.552 | -3.304 |
| CLPF-Independent | 0.800 | -0.326 | 4.970 | 7.924 |
| CLPF-Wiener | 0.390 | -0.790 | 1.041 | -1.885 |
| Latent SDE | 1.243 | 0.082 | 3.594 | 7.740 |
| CLPF | 0.444 | -0.831 | 1.322 | -2.620 |

Table 4: **Sequential Prediction (Real-World Data).** We report the average L2 distance between prediction results and ground truth observations in a sequential prediction setting. The prediction is based on the average of 125 samples. Results are reported in the format *mean, [25th percentile, 75th percentile]*.

| Model | Mujoco [32] | BAQD [35] | PTBDB [4] |
|---|---|---|---|
| VRNN [9] | 1.599, [0.196, 1.221] | 0.519, [0.168, 0.681] | 0.037, [0.005, 0.032] |
| Latent ODE [32] | 13.959, [9.857, 15.673] | 1.416, [0.936, 1.731] | 0.224, [0.114, 0.322] |
| Latent SDE [24] | 7.627, [2.384, 8.381] | 0.848, [0.454, 1.042] | 0.092, [0.032, 0.111] |
| CTFP [11] | 1.969, [0.173, 1.826] | 0.694, [0.202, 0.966] | 0.055, [0.006, 0.046] |
| Latent CTFP [11] | 1.983, [0.167, 1.744] | 0.680, [0.189, 0.943] | 0.065, [0.007, 0.059] |
| CLPF-ANODE (**ours**) | 1.629, [0.149, 1.575] | 0.542, [0.150, 0.726] | 0.048, [0.005, 0.041] |
| CLPF-iRes (**ours**) | 1.846, [0.177, 1.685] | 0.582, [0.183, 0.805] | 0.055, [0.006, 0.049] |

## 5.2 Real-world Data

Many real-world systems, despite having continuous dynamics, are recorded via observations at a fixed sampling rate. Mujoco-Hopper [32], Beijing Air-Quality Dataset (BAQD) [35] and PTB Diagnostic Database (PTBDB) [4, 16] are examples of such datasets. We create training/testing data at irregular times by drawing time stamps from a Poisson process and mapping them to the nearest observed sample points (see Appendix D for details).

We demonstrate our model's ability to fit this type of observation as a surrogate for its ability to capture real-world continuous dynamics. Additionally, we also evaluate CLPF on a sequential prediction task to illustrate its benefits in an online inference setting. The results for these experiments are reported in Table 2 and Table 4. In the likelihood estimation task our model achieves competitive performance with respect to other methods that generate continuous dynamics. We do observe low negative log-likelihoods (NLLs) by VRNN on these data, though note that VRNN does not generate true continuous dynamics. In the sequential prediction task CLPF outperforms the other baselines with continuous dynamics; the performance gap between CLPF and VRNN is significantly reduced compared to likelihood estimation.

In the second row of Fig. 3, we also visualize one dimension of sample sequences generated from CLPF and VRNN models trained on a dense time grid of BAQD data (gap between steps: 0.01) and compare with ground truth samples from the dataset. We can see that VRNN fails to generate trajectories visually similar to samples from the ground truth data on a dense time grid, despite its favourable NLL on a sparse and irregular time grid. The sampled trajectories tend to converge and show little variance after running VRNN for a large number of iterations (1500 for $t = 1.5$). We also observe that VRNN fails to generate trajectories with consistent visual properties when we sample observations on different time grids (see supplementary material for more qualitative samples).

## 5.3 Ablation Study

We compare the proposed CLPF models against three variants that are obtained by making changes to key components of CLPF. The results with these models are shown in Table 3. **CLPF-Global** is a variant created by replacing our piecewise construction with a global posterior process similar to latent SDE [24]. **CLPF-Independent** is a variant where the continuously indexed normalizing flows

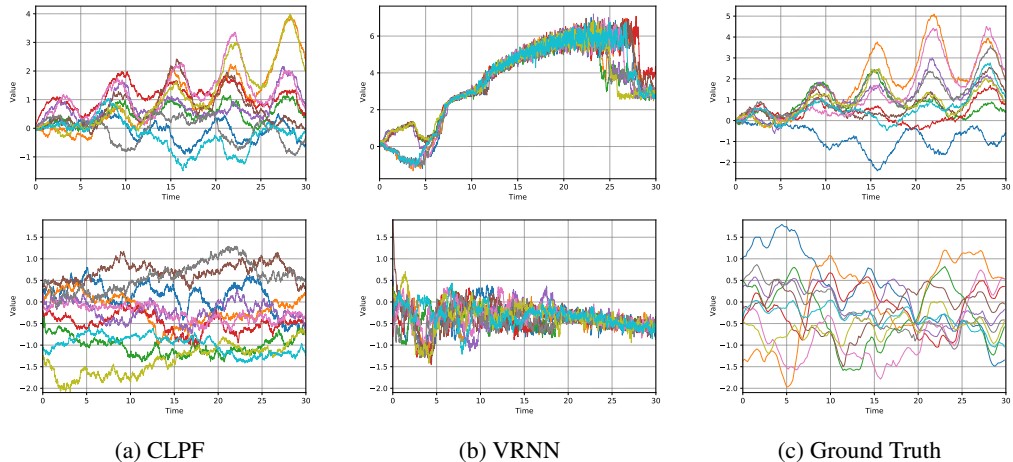

|                |                |                |
|:--------------:|:--------------:|:--------------:|
| (a) CLPF       | (b) VRNN       | (c) Ground Truth |

Figure 3: **Qualitative Samples on Dense Time Grid of CLPF and VRNN Models.** We generate the sample sequences from CLPF and VRNN trained with LSDE (first row) and BAQD (second row) data and compare the samples generated from the model to samples from the ground truth process (LSDE) or training set (BAQD).

and CTFP-decoding are replaced with static distributions and independent decoders. **CLPF-Wiener** is a variant that uses a Wiener process as the base process for the CTFP decoder.

CLPF-Global differs from CLPF in a manner similar to the difference between latent SDE and CLPF-Independent. In both comparisons we can see modest improvements of performance when we replace the posterior process defined by a single SDE with a piecewise constructed posterior. Both the comparisons between CLPF and CLPF-Independent and between CLPF-Global and latent SDE show that a CTFP-style decoding into continuous trajectories is indeed a better inductive bias when modeling data with continuous dynamics. Although the comparison with CLPF-Wiener does not show a clear trend, CLPF still delivers an overall better performance on the synthetic datasets. Therefore we use the OU process as our base process in all CLPF experiments.

## 6 Discussion and Conclusion

**Limits.** While being more expressive than the CTFP model, the proposed CLPF model is not capable of exact likelihood evaluation and relies on variational approximations for estimation and optimization. The CLPF model can also comprise up to two differential equations that need to be numerically solved, potentially incurring higher computational cost. Fortunately, this computational cost can be controlled via additional parameters: (1) The approximation error of numerical ODE/SDE solutions can be controlled through the tolerance parameters of numerical solvers and the inversion error of residual flows can be controlled through the number of fixed-point iterations. [3]; (2) The tightness of the IWAE bound for likelihood estimation can be controlled through the number of latent samples.

**Broader Impact.** As an expressive generative model for continuous time-series, we believe CLPF can be used to model continuous dynamics from partial observations in a wide range of areas, including physics, healthcare, and finance. Furthermore, it can potentially be leveraged in a variety of downstream tasks, including data generation, forecasting, and interpolation. However, care must be taken that CLPF models are not used for adverse inference on restricted data. Protection of critical unobserved information is an active area of research [1] that can be explored as an orthogonal component in future versions of our model.

**Conclusion.** We have presented Continuous Latent Process Flows (CLPF), a generative model of continuous dynamics that enables inference on arbitrary real time grids, a complex operation for which we have also introduced a powerful piecewise variational approximation. Our architecture is built around the representation power of a flexible stochastic differential equation driving a continuously indexed normalizing flow. An ablation study as well as a comparison to state-of-the-art baselines for continuous dynamics have demonstrated the effectiveness of our contributions on both synthetic and real-world datasets. A set of qualitative results support our findings. In the future, we plan to explore the theoretical properties of our model in more detail, including an analysis of its universal approximation properties and its ability to capture non-stationary dynamics.

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
