# Supplementary Materials for Continuous Latent Process Flows

## A    Stochastic Processes Representable by Continuous Time Flow Processes

We characterize the class of stochastic processes that CTFP [7] can model and show that it does not include the widely used Ornstein-Uhlenbeck (OU) process. To this end, we first express the CTFP deformation of a Wiener as the solution of a stochastic differential equation (Lemma 1) and then demonstrate that this representation cannot match the drift term of an OU process (Theorem 1).

**Lemma 1.** *Let $\boldsymbol{W}_t$ be a Wiener process defined on a filtered probability space $(\boldsymbol{\Omega}, \boldsymbol{\mathcal{F}}_t, P)$ for $0 \leqslant t \leqslant T$ and $f(\boldsymbol{w}, t)$ be a function satisfying the following conditions:*

- *$f(\boldsymbol{w}, t)$ is twice continuously differentiable on $\mathbb{R} \times [0, T]$;*

- *for each $t$, $f(\boldsymbol{w}, t) : \mathbb{R} \to \mathbb{R}$ is bijective and Lipschitz-continuous w.r.t. $\boldsymbol{w}$.*

*Let $f^{-1}(\cdot, t)$ denote the inverse of $f(\cdot, t)$. The stochastic process $\boldsymbol{X}_t = f(\boldsymbol{W}_t, t)$ is a solution to the stochastic differential equation (SDE)*

$$d\boldsymbol{X}_t = \{\frac{df}{dt}(f^{-1}(\boldsymbol{X}_t, t), t) + \frac{1}{2}\operatorname{Tr}(\mathbf{H}_{\boldsymbol{w}}f(f^{-1}(\boldsymbol{X}_t, t), t))\}\, dt + (\nabla_{\boldsymbol{w}}f^T(f^{-1}(\boldsymbol{X}_t, t), t))\, d\boldsymbol{W}_t, \quad (1)$$

*with initial value $\boldsymbol{X}_0 = f(0, \mathbf{0})$.*

*Proof.* Applying Itô's Lemma to $\boldsymbol{X}_t = f(\boldsymbol{W}_t, t)$, we obtain

$$\mathrm{d}f(\boldsymbol{W}_t, t) = \{\frac{\mathrm{d}f}{\mathrm{d}t}(\boldsymbol{W}_t, t) + \frac{1}{2}\operatorname{Tr}(\mathbf{H}_{\boldsymbol{w}}f(\boldsymbol{W}_t, t))\}\, \mathrm{d}t + (\nabla_{\boldsymbol{w}}f^T(\boldsymbol{W}_t, t))^T\, \mathrm{d}\boldsymbol{W}_t, \quad (2)$$

where $\mathbf{H}_{\boldsymbol{w}}f(\boldsymbol{W}_t, t)$ is the Hessian matrix of $f$ with respect to $\boldsymbol{w}$ and $\frac{\mathrm{d}f}{\mathrm{d}t}(\boldsymbol{W}_t, t)$ and $\nabla_{\boldsymbol{w}}f(\boldsymbol{W}_t, t)$ are derivatives with respect to $t$ and $\boldsymbol{w}$, all evaluated at $\boldsymbol{W}_t, t$. Since $f(\boldsymbol{w}, t)$ is invertible in $\boldsymbol{w}$ for each $t$, we have $\boldsymbol{W}_t = f^{-1}(\boldsymbol{X}_t, t)$ and can rewrite Eq.(2) as

$$\mathrm{d}\boldsymbol{X}_t = \{\frac{\mathrm{d}f}{\mathrm{d}t}(f^{-1}(\boldsymbol{X}_t, t), t) + \frac{1}{2}\operatorname{Tr}(\mathbf{H}_{\boldsymbol{w}}f(f^{-1}(\boldsymbol{X}_t, t), t))\}\, \mathrm{d}t + (\nabla_{\boldsymbol{w}}f^T(f^{-1}(\boldsymbol{X}_t, t), t))\, \mathrm{d}\boldsymbol{W}_t. \quad (3)$$

$\square$

**Theorem 1.** *Given a one-dimensional Wiener process $\boldsymbol{W}_t$ on a filtered probability space $(\boldsymbol{\Omega}, \boldsymbol{\mathcal{F}}_t, P)$ for $0 \leqslant t \leqslant T$, a stochastic process $\boldsymbol{X}_t = f(\boldsymbol{W}_t, t)$ with $f$ satisfying the conditions of Lemma 1 cannot be the (strong) solution to the SDE of a one-dimensional Ornstein–Uhlenbeck (OU) process $d\boldsymbol{X}_t = -\theta\boldsymbol{X}_t\, dt + \sigma d\boldsymbol{W}_t$ with the initial condition $\boldsymbol{X}_0 = f(0, \mathbf{0})$ for any constants $\theta > 0$ and $\sigma > 0$.*

*Proof.* We can obtain the stochastic differential equation of $\boldsymbol{X}_t = f(\boldsymbol{W}_t, t)$ using Lemma 1. For $\boldsymbol{X}_t = f(\boldsymbol{W}_t, t)$ to be the strong solution of an *OU process*, it must satisfy

$$\begin{aligned}
\boldsymbol{X}_t &= \boldsymbol{X}_0 + \int_0^t -\theta\boldsymbol{X}_s\, \mathrm{d}s + \int_0^t \sigma\, \mathrm{d}\boldsymbol{W}_s \\
&= \boldsymbol{X}_0 + \int_0^t \frac{\mathrm{d}f}{\mathrm{d}t}(f^{-1}(\boldsymbol{X}_s, s), s) + \frac{1}{2}\frac{\mathrm{d}^2f}{\mathrm{d}\boldsymbol{w}^2}(f^{-1}(\boldsymbol{X}_s, s), s)\, \mathrm{d}s + \int_0^t \frac{\mathrm{d}f}{\mathrm{d}\boldsymbol{w}}(f^{-1}(\boldsymbol{X}_s, s), s))\, \mathrm{d}\boldsymbol{W}_s
\end{aligned}$$

$$(4)$$

almost surely. In particular, this implies that the drift and variance terms of $\boldsymbol{X}_t$ must match the drift and variance terms of the OU process. We have $\forall (\boldsymbol{w}, t) \in \mathbb{R} \times [0, T] : \frac{\mathrm{d}f}{\mathrm{d}\boldsymbol{w}}(\boldsymbol{w}, t) = \sigma$. As a result, $f(\boldsymbol{w}, t)$ must take the form $\sigma \cdot \boldsymbol{w} + g(t)$ for some function $g(t)$ of $t$. However, this form of $f(\boldsymbol{w}, t)$ implies the value of $\frac{\mathrm{d}f}{\mathrm{d}t}(\boldsymbol{w}, t) + \frac{1}{2}\frac{\mathrm{d}^2 f}{\mathrm{d}\boldsymbol{w}^2}(\boldsymbol{w}, t)$ is independent of $\boldsymbol{w}$ as $\frac{\mathrm{d}f}{\mathrm{d}t}$ is a function of $t$ and $\frac{1}{2}\frac{\mathrm{d}^2 f}{\mathrm{d}\boldsymbol{w}^2}$ is a constant. Therefore the drift term of the SDE of $\boldsymbol{X}_t$ cannot match the drift term of the OU process. $\qquad\square$

## B  Lipschitz Constraints in CTFP

We use the following example to qualitatively illustrate the challenges implied by the Lipschitz constraints of some popular normalizing flow models if we attempt to directly use them in CTFP [7] to transform a Wiener process $\boldsymbol{W}_t$ to a geometric Brownian motion. Let $\boldsymbol{W}_{t_i}$ be the marginal random variable induced by the base process $\boldsymbol{W}_t$ at time point $t_i$ which follows a Gaussian distributions with parameters $\mu$ and $\sigma$ and $\boldsymbol{X}_{t_i}$ be the random variable induced by the target GBM at time point $t_i$ which follows a log-normal distribution with parameters $\mu'$ and $\sigma'$. Without loss of generality we assume $\mu = \mu' = 0$ and $\sigma = \sigma' = 1$. Consider a bi-lipschitz invertible mapping $f : \mathbb{R} \rightarrow \mathbb{R}$. By the change of variable formula we have $\log \boldsymbol{p}(f(\boldsymbol{W}_{t_i})) = \log \boldsymbol{p}(\boldsymbol{W}_{t_i}) - \log \left| \frac{\partial f(\boldsymbol{W}_{t_i})}{\partial \boldsymbol{W}_{t_i}} \right| = -\frac{\boldsymbol{W}_{t_i}^2}{2} - \log \left| \frac{\partial f(\boldsymbol{W}_{t_i})}{\partial \boldsymbol{W}_{t_i}} \right| +$ constant. $\log \boldsymbol{p}(f(\boldsymbol{W}_{t_i}))$ decreases at a rate of $\mathcal{O}(\boldsymbol{W}_{t_i}^2)$ when $\boldsymbol{W}_{t_i} \rightarrow \infty$ as $\log \left| \frac{\partial f(\boldsymbol{W}_{t_i})}{\partial \boldsymbol{W}_{t_i}} \right|$ is bounded from both above and below. Directly evaluating $f(\boldsymbol{W}_{t_i})$ using the density function of of log-normal distribution $\tilde{p}$, we get $\log \tilde{p}(f(\boldsymbol{W}_{t_i})) = -f(\boldsymbol{W}_{t_i}) - \frac{(\log \boldsymbol{W}_{t_i})^2}{2} +$ constant which decreases at the rate of $\mathcal{O}(\boldsymbol{W}_{t_i} + (\log \boldsymbol{W}_{t_i})^2)$. This comparison indicates the tail behaviour of $f(\boldsymbol{W}_{t_i})$ can not match that of a log normal random variable as $\boldsymbol{W}_{t_i}$ tends to infinity.

Many popular normalizing flow implementations that can be used as basic building blocks of CTFP including residual flows [4, 2] and neural ODE [3] are bi-lipschitz mappings by design [10].

## C  Synthetic Dataset Specifications

We compare our models against the baseline models using data simulated from four continuous stochastic processes: geometric Brownian motion (GBM), linear SDE (LSDE), continuous auto-regressive process (CAR), and stochastic Lorenz curve (SLC). We simulate the observations of GBM, LSDE, and CAR in the time interval $[0, 30]$ and the observations of SLC in the time interval $[0, 2]$. For each trajectory, we sample the observation time stamps from an independent homogeneous Poisson process with intensity 2 (i.e., the average interarrival time of observations is 0.5) for GBM, LSDE, and CAR and an intensity of 20 (i.e., the average inter-arrival time of observations is 0.05) for SLC. The observation values for geometric Brownian motion are sampled according to the exact transition density. The observation values of the LSDE, CAR, and SLC processes are simulated using the Euler-Maruyama method [1] with a step size of $1e - 5$. For each process, we simulate 10000 sequences, of which 7000 are used for training, 1000 are used for validation and 2000 are used for evaluation. We further simulated 2000 sequences with denser observations using an intensity of 20 for GBM, LSDE, and CAR and an intensity of 40 for SLC.

In the remainder of this section we provide details about the parameters of the stochastic processes:

**Geometric Brownian Motion.** The stochastic process can be represented by the stochastic differential equation $\mathrm{d}\boldsymbol{X}_t = 0.2\boldsymbol{X}_t \,\mathrm{d}t + 0.1\boldsymbol{X}_t \,\mathrm{d}\boldsymbol{W}_t$, with an initial value $\boldsymbol{X}_0 = 1$.

**Linear SDE.** The linear SDE we simulated has the form $\mathrm{d}\boldsymbol{X}_t = (0.5 \sin(t)\boldsymbol{X}_t + 0.5\cos(t)) \,\mathrm{d}t + \frac{0.2}{1+\exp(-t)} \,\mathrm{d}\boldsymbol{W}_t$. The initial value was set to 0.

**Continuous AR(4) Process.** A CAR process $\boldsymbol{X}_t$ can be obtained by projecting a high-dimensional process to a low dimension. This process tests our model's ability to capture non-Markov processes:

$$
\begin{aligned}
\boldsymbol{X}_t &= [1, 0, 0, 0]\boldsymbol{Y}_t, \\
\mathrm{d}\boldsymbol{Y}_t &= A\boldsymbol{Y}_t \,\mathrm{d}t + e \,\mathrm{d}\boldsymbol{W}_t,
\end{aligned}
\quad \text{where } A = \begin{bmatrix} 0 & 1 & 0 & 0 \\ 0 & 0 & 1 & 0 \\ 0 & 0 & 0 & 1 \\ a_1 & a_2 & a_3 & a_4 \end{bmatrix},
\tag{5}
$$

$$e = [0, 0, 0, 1], \ [a_1, a_2, a_3, a_4] = [+0.002, +0.005, -0.003, -0.002].$$

**Stochastic Lorenz Curve.** A stochastic Lorenz curve is a three-dimensional stochastic process that can be obtained by solving the stochastic differential equations

$$
\begin{aligned}
\mathrm{d}\boldsymbol{X}_t &= \sigma(\boldsymbol{Y}_t - \boldsymbol{X}_t)\,\mathrm{d}t + \alpha_x\,\mathrm{d}\boldsymbol{W}_t, \\
\mathrm{d}\boldsymbol{Y}_t &= (\boldsymbol{X}_t(\rho - \boldsymbol{Z}_t) - \boldsymbol{Y}_t)\,\mathrm{d}t + \alpha_y\,\mathrm{d}\boldsymbol{W}_t, \\
\mathrm{d}\boldsymbol{Z}_t &= (\boldsymbol{X}_t\boldsymbol{Y}_t - \beta\boldsymbol{Z}_t)\,\mathrm{d}t + \alpha_z\,\mathrm{d}\boldsymbol{W}_t.
\end{aligned}
\tag{6}
$$

In our experiments we use $\sigma = 10$, $\rho = 28$, $\beta = 8/3$, $\alpha_x = 0.1$, $\alpha_y = 0.28$, and $\alpha_z = 0.3$.

# D  Real-World Dataset Specifications

We use three real-world datasets to evaluate our model and the baseline models. We follow a similar data preprocessing protocol as [7], which creates step functions of observation values in a continuous time interval from synchronous time series data. The sequences in each dataset are padded to the same length: 200 for Mujoco-Hopper, 168 for Beijing Air Quality Dataset (BAQD), and 650 for PTB Diagnostic Database (PTBDB). The indices of observations are treated as real numbers and rescaled to a continuous time interval: for Mujoco-Hopper and BAQD we use the time interval $[0, 30]$, for PTBDB we use the time interval $[0, 120]$. We sample irregular observation time stamps from a homogeneous Poisson process of intensity 2. For each sampled time stamp, we use the observation value of the closest available time stamp (rescaled from indices). As the baseline CTFP model [7] makes a deterministic prediction at $t = 0$, we follow their steps and shift the sampled observation time stamps by $0.2$ after obtaining their corresponding observation values.

# E  Model Architectures and Experiment Settings

We keep the key hyperparameters of our model and the baseline models in a similar range, including the dimensions of the recurrent neural networks' hidden states, the dimensions of latent states, and the hidden dimensions of decoders. We set the latent dimension to 2 for geometric Brownian motion and linear SDE, 4 for the continuous auto-regressive process, 3 for the stochastic Lorenz curve, and 64 for all real-world datasets. For all models that use a recurrent neural network, we use gated recurrent units with a hidden state of size 16 for the synthetic datasets and 128 for the real-world datasets.

**Continuous Latent Process Flows (CLPF).** We use a fully-connected network with two hidden layers to implement the drift $\boldsymbol{\mu}$ both for the prior and posterior SDE. To implement the variance function $\boldsymbol{\sigma}$, We use additive noise for experiments on synthetic datasets and a network with the same architecture as the drift $\boldsymbol{\mu}$ for real-world data experiments. The hidden layer dimensions for (prior SDE, posterior SDE) are $(32, 32)$ for the synthetic datasets and $(128, 64)$ for the real-world datasets. We use a Gated Recurrent Unit (GRU) as the encoder of observation $\boldsymbol{x}_{t_i}$s and latent states $\boldsymbol{z}_{t_i}$s to produce $\phi_i$ at each step $i$ in Eq.(13) in the main paper. The GRU takes the observation $\boldsymbol{X}_{t_i}$, the latent state $\boldsymbol{Z}_{t_i}$, the current and previous time stamps $t_i$ and $t_{i-1}$, and the difference between the two time stamps as inputs. The updated hidden state is projected to a context vector of size 16 for the synthetic datasets and 20 for the real-world datasets. The projected vector is concatenated with $\boldsymbol{X}_{t_i}$ and $t_i$ as part of the input to the drift function $\boldsymbol{\mu}_{\phi_i}$ of the posterior process in the interval $[t_{i-1}, t_i]$.

We use five blocks of the generative variant of augmented neural ODE (ANODE) [7] or indexed residual flows [6] to implement the indexed normalizing flows in all synthetic and real-world experiments. In each ANODE block, the function $\boldsymbol{h}$ in Eq.(7) in the main paper is implemented as a neural network with 4 hidden layers of dimension $[8, 32, 32, 8]$ for the synthetic datasets and $[16, 32, 32, 16]$ for the real-world datasets; the funtion $g$ is implemented as a zero mapping. Indexed residual flows are only used in experiments on real-world data. Each residual flow block $f_i$ in Eq.(8) in the main paper is implemented using networks with the same hidden dimensions as ANODE for the real-world experiments. $u^{(i)}$ and $v^{(i)}$ for each block are obtained by projection of the latent state vector using a fully-connected network with 2 hidden layers of dimension $[32, 32]$. Please refer to the code base for more details.

**Continuous Time Flow Process (CTFP) and Latent CTFP.** For CTFP [7] and the decoder of its latent variant, we also use 5 ANODE blocks with the same number of hidden dimensions as CLPF. The encoder of the latent CTFP model is an ODE-RNN [9]. The ODE-RNN model consists of a

recurrent neural network and a neural ODE module implemented by a network with one hidden layer of dimension 100. The default values in the official implementation[1] of latent CTFP are adopted for other hyperparameters of the model architecture.

**Latent ODE.**   For the latent ODE model [9], we use the same encoder as latent CTFP. The latent ODE decoder uses a neural ODE with one hidden layer of dimension 100 to propagate the latent state across a time interval deterministically. The latent state propagated to each observation time stamp is mapped to the mean and variance of a Gaussian observational distribution by a fully-connected network with 4 hidden layers of dimension $[16, 64, 64, 16]$ for synthetic datasets and $[32, 64, 64, 32]$ for the real-world datasets. We use the default values in the official implementation[2] of latent ODE for other hyperparameters of the model architecture.

**Latent SDE.**   In our implementation of latent SDE [8], we use the same architectures for the drift function $\mu$ and variance function $\sigma$ as CLPF. A GRU with the same hidden dimension is used to encode observation sequence $x_{t_i}$s and only outputs a single set of parameter $\phi$ for the drift function of the variational posterior process. We use the same architecture for the decoder from latent states to observational distributions as latent ODE.

**Variational RNN (VRNN).**   The backbone of VRNN [5] is a recurrent neural network. A one-layer GRU is used to implement the recurrent neural network. During inference, the hidden state is projected to the mean and variance of a Gaussian distribution of the latent state by a fully-connected network. During generation, the hidden state is directly mapped to the parameters of the latent distribution. The sample of the latent state is decoded to the parameters of an observational Gaussian distribution by a fully-connected network with 4 hidden layers of dimension $[16, 64, 64, 16]$ for the synthetic datasets and $[32, 64, 64, 32]$ for the real-world datasets. In the recurrence operation, the GRU takes the latest latent sample and observation as inputs. We also concatenate the time stamp of the current observation as well as the difference between the time stamps of the current and previous observation to the input.

**Experiment Settings.**   For each dataset, we use $70\%$ of the sequences for training, $10\%$ of the sequences for validation, and $20\%$ of the sequences for testing. For the real-word datasets, we add Gaussian noise with standard deviation $1e-3$ to the training data to stabilize the training. We use a batch size of 100 for training on synthetic data and a batch size of 25 for the real-world datasets. For models optimized with an IWAE bound, we use 3 samples of the latent state (trajectory) for training on synthetic datasets, 5 samples for training on real-world datasets, 25 samples for validation, and 125 samples for evaluation. To solve the latent stochastic process in the continuous latent process flow model, we use the Euler-Maruyama scheme with adaptive step size. The automatic differentiation engine of PyTorch is used for backpropagation. We train our models using one GPU (P100 / GTX 1080ti) on all datasets except PTBDB which uses 2 GPUs for training. The training takes approximately 80 hours for each synthetic dataset, except for CAR process which requires twice the time for training. The training time for real-world datasets is approximately 170 hours.

**Qualitative Samples**   We presents more qualitative samples generated by latent ODE [9], latent SDE [8], CTFP [7], and latent CTFP on a dense time grids with step gap of 0.01 as well as samples from VRNN models generated on time grids with step gaps of 0.2 amd 0.5 in Fig. 1. The models are trained on BAQD [11] datasets.

---

[1]https://github.com/BorealisAI/continuous-time-flow-process
[2]https://github.com/YuliaRubanova/latent_ode

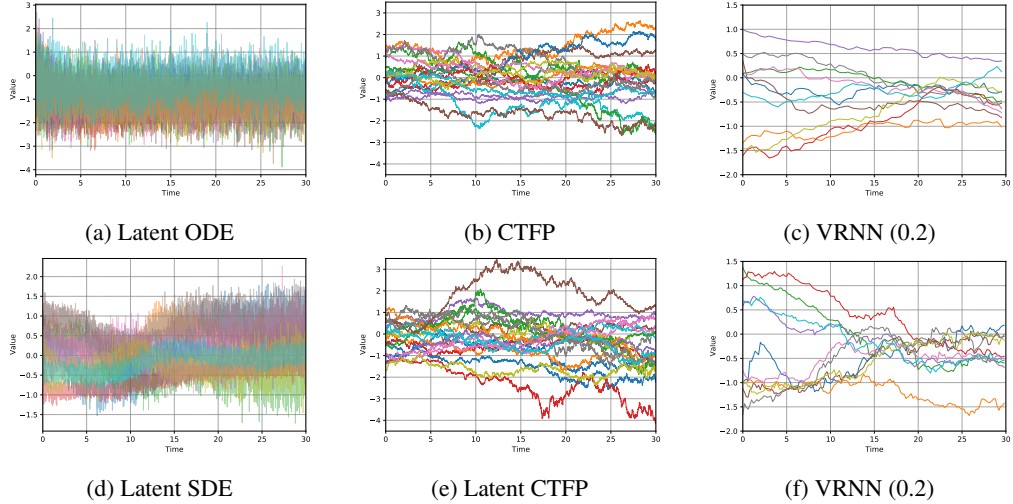

(a) Latent ODE            (b) CTFP            (c) VRNN (0.2)

(d) Latent SDE            (e) Latent CTFP            (f) VRNN (0.2)

Figure 1: Qualitative samples from latent ODE, latent SDE, CTFP and latent CTFP on dense time grids with gap of 0.01 and VRNN Models on time grids with different gaps. The numbers in the parentheses in the captions of Fig. 1c and Fig. 1f indicate the gap of the time grids on which we sample observations from VRNN models.