# OpenReview forum: "Continuous Latent Process Flows"
_NeurIPS.cc/2021/Conference — NeurIPS 2021 Poster_

### Official Review · Reviewer_GfLk · 2021-06-28

**Rating:** 4
**Confidence:** 5

**Summary:**

The paper introduces a new form of generative model for continuous time-series consisting of a latent neural SDE transformed by a continuously indexed normalizing flow. This is motivated by the fact that continuously indexed normalizing flows with latent Brownian motions can only model a restricted class of SDE dynamics while neural SDE can only model Markov processes.

**Limitations And Societal Impact:**

The limitations of their approach are only briefly discussed.

**Main Review:**

The paper addresses with the important problem of modeling discretely sampled continuous dynamics. The work is a development on recent work on continuous normalizing flows. Specifically, it can be seen as an improvement of the continuous-time flow processes [1].

The main difference between [1] and the current work is the use of a latent neural SDE instead of a the Wiener process as latent process. This greatly increases the representational flexibility of the approach since many Ito processes cannot be obtained by transforming a Weiner process with a bijector. All in all, the approach is natural improvement of [1] which fixes a clearly identifiable problem. The resulting method is however less elegant as it combines two separate parameterized components, each of which would work as stand-alone generative model. The novelty is limited since the approach is a natural combination of two popular methods without key new insights and model-specific derivations.  The paper also introduces a somehow novel variational inference method to infer the latent form the discretely sampled observations.

Strengths:
- The paper addresses a clearly defined problem with important applications. It introduces a new method that solves some clearly stated limitation of state-of-the-art techniques.

Weaknesses:
- The novelty of the approach is rather limited, being the direct combination of two existing methods.
-The experiments are on the weak side for NeurIPS. I would have liked to see more experiments both on simulated timeseries and on real data. Given the setting, it would have also be informative to compare the result with Gaussian process and neural process baselines. The results are difficult to interpret since the paper does not provide error bars. The results on real-world data are rather underwhelming, especially for the CLPF-ANODE version, which is the one described in the methods section. There is a big difference between the continuous approaches and the much simpler VRNN in terms of likelihood. The proposed method does not seem to perform much better than the much simpler CTFP.

[1] Deng, Ruizhi, et al. "Modeling continuous stochastic processes with dynamic normalizing flows." 34th Conference on Neural Information Processing Systems



**Time Spent Reviewing:**

3

---

> ### Author Response · Authors · 2021-08-11
> **Response to Reviewer GfLk**
>
> We would like to thank the reviewer for recognizing the importance of the problem area that our work focuses on and the novelty of the proposed variational inference methods. We would like to address the reviewer’s concerns as follows:
>
> **Difference between CTFP and CLPF** [“The main difference between [1] and the current work...”]
> We believe there is a critical misunderstanding: the major difference between our work and CTFP [1] is **not** the replacement of the Wiener process with a latent SDE but to **drive** the CTFP model with a latent SDE. Furthermore, Section 3.1 discusses the advantages of replacing the base Wiener process in the CTFP model by an OU process; see Table 2 for an experimental comparison. Note that Reviewer XbGb proposed an alternative approach similar to the reviewer’s understanding of our work; please refer to [Reviewer XbGb, Importance of $Z_t$ and $O_t$] for a more detailed discussion on the challenges of this approach.
>
> We would add this discussion to a revised version to improve the clarity of the difference between our work and CTFP [1].
>
> **Novelty** [“The novelty of the approach is rather limited”]
> The originality of our work is driving indexed normalizing flows using a latent SDE to warp stochastic processes.  The identification of theoretical drawbacks in previous models and the proposed variational inference method are also major contributions in the work and motivate the improvements. Apart from solving the identified problems of existing methods, please refer to [Importance of $Z_t$ and $O_t$] in our response to Reviewer XbGb for an in-depth discussion on the reasoning and importance of combining these two methods into our model. We hope this response helps to clarify the novelty and contributions of the work.
>
> **Experiments** [“The experiments are on the weak side for NeurIPS”]
>
> In terms of experiment results, we believe the purest way of exposing the drawbacks of existing methods and illustrating the benefits of CLPF is to evaluate the model on synthetic data directly sampled from common stochastic processes using negative log-likelihood. The experiments on synthetic data are carefully designed and the results validate our motivation and the significance of CLPF.
>
> The reviewer suggested comparing with other baseline models including Gaussian processes and neural processes. We feel it is necessary to point out a critical difference between the models presented in the experiment section and many models in the neural process family: while CLPF, CTFP, latent ODE, and latent SDE are generic generative models, many neural processes directly model conditional distributions given contextual observations.
>
> The experimental results on real-world datasets can be impacted by a lot of factors, including data preprocessing and hyper-parameter tuning. CLPF still shows competitive (and in many cases superior) performance in a comparison against baseline models capable of learning continuous dynamics, including CTFP, latent CTFP, latent SDE, and latent ODE.
>
> We will provide additional experiment results evaluated using RMSE and error bars of the experimental results in a separate post.
>
> [1] Deng, Ruizhi, et al. "Modeling continuous stochastic processes with dynamic normalizing flows." 34th Conference on Neural Information Processing Systems

---

> > ### Comment · Reviewer_GfLk · 2021-08-12
> > **Response**
> >
> > Dear authors,
> >
> > Thank you for the thorough response. I do acknowledge that I misunderstood a part of the method, however it is far from being a critical misunderstanding since, as far as I understand it, driving CTFP with a neural SDE is equivalent to replacing the Weiner process with a (integrated) neural SDE.
> >
> > So said, I do agree that identifying a major limitation of CTFP and fixing it is a substantial contribution and worth of NeurIPS acceptance. However, the empirical results remain rather underwhelming when compared with the parametric baseline and the range and selection of baselines is too limited. I do think tthat this paper has potential but I am not sure that it is ready for publication. However, I will not appose acceptance if all other reviewers vote in favor.

---

> > ### Author Response · Authors · 2021-08-25
> > **Response Updates to Reviewer GfLk**
> >
> > We would like to thank the reviewer for the encouraging feedback.
> >
> > **Experiments** [“The experiments are on the weak side for NeurIPS”]
> >
> > While analyzing the differences in real-world performance between VRNN and CLPF we discovered inconsistency in our experimental protocols that gives an advantage to VRNN. Aligning the protocols results in significant performance improvements of our model and almost completely eliminates the gap to VRNN. Code to reproduce the experiment and its results will be published together with the paper. We also evaluated the models using root mean square error (RMSE) as the metric. Please refer to **Experimental Evaluation** in **Response Updates to Reviewer 1ACw** for the experiment results. We hope the updated experiment results can further address the reviewer’s concerns.

---

### Official Review · Reviewer_1ACw · 2021-07-05

**Rating:** 4
**Confidence:** 3

**Summary:**

The paper addresses probabilistic modeling of sparse or irregularly sampled time series. The proposed method captures the systems dynamics by a rather simple latent stochastic process, which is then warped into a more complex one using a time-indexed ANODE-like normalizing flow. The free parameters of the dynamical system are identified by maximizing the ELBO that lower bounds the likelihood of the observed sequence. The solution aims to overcome the limited expressive power of normalizing flows stemming for the necessity to use invertible transformations that also need to satisfy Lipschitz continuity restrictions.

**Ethical Concerns:**

This is a generic methods development paper. Hence I do not see any identifiable ethical concerns specific to the problem it addresses or the solutions it proposes. However, I would still expect the paper to provide more thoughts on this somewhere in the manuscript. The answers provided in the checklist on this issue are too generic and superficial.

**Limitations And Societal Impact:**

I do not find the answer of the authors to the checklist question 1 (b) convincing. Having closely read Section 3, I am not able to point out a clear discussion anywhere in the paper about the limitations of the "original solution" proposed by this work. The paper answers to some extent previously unsolved problem this paper has solved, but does not really talk about what its leaves unsolved or at the expense of what the solution has been achieved (e.g. increased computational cost, additional assumptions on the latent space, loss of interpretability etc.).

**Main Review:**

Significance of Novelty
--

A key design choice is to assume the latent dynamics to be governed by an Ornstein Uhlenbeck process, which is a rather trivial stochastic process with analytical solution. The beauty of the solution comes when this simple process is warped by a normalizing flow into dynamical models with arbitrary complexity. This is a compelling solution, and however simple, a novel use of stochastic differential equations in modern machine learning. The suggested variational inference algorithm, especially piecewise reweighting part is also interesting and novel for the context in question. However, there do exist alternative solutions to all the problems pointed out in the paper, which appear to work in the reported experiments better than the proposed method. The lack of empirical evidence on the benefit of the approach impairs its significance.

Presentation
--
Section 2 gives a really nice, concice, and clear overview into the essential material to be used later. It is remarkable how a variety of complex concepts are introduced in such clear terms. However, I find the structure of Section 3 a bit convoluted. It takes a while to grasp how the pieces come together in the second and third paragraphs of Section 3.1. Further, the novelty claims are also a bit hidden. I recommend that the next revision of the paper includes a dedicated paragraph that explain in technical depth what exactly is different in the proposed solution from the state of the art. Lastly, presenting the whole method also as pseudocode in the main paper as Algorithm 1 would substantially improve the reader experience.

Experimental Evaluation
--

Both the baselines and the experiment setups are well-chosen and well-justified. However, the same is not the case for the evaluation metrics. The paper reports only the test log-likelihood in all experiments. While obviously being a useful score, it is not sufficient to draw conclusions about the prediction success by looking only at the log likelihood. The authors should definitely provide the plain RMSE scores of the same experiments in the rebuttal phase. Furthermore, the proposed method does not seem to improve the evaluated performance metric in the real-world data sets with respect to existing alternative solutions. It could well be that the proposed method has some properties more favorable than its competitors beyond prediction accuracy, but then there should be a corresponding experiment or evaluation metric that quantifies this favorableness.

Questions:
--

Fig 2 shows that in the data generating process, X_t depends on both Z_t and Z_{t-1}. Where does the dependency on Z_{t-1} come from? Which equation in the paper clarifies this dependency? Eq 6 is a non-Markoving mapping.

Overall
--

Interesting problem selection. The solution has some not groundbreaking but fairly novel aspects. Scientific writing is overall OK, but the novelty presentation (Sec 3) could be improved. Experimental evaluation is extremely weak. The solution falls behind multiple baselines. Evaluation wrt RMSE missing.

**Time Spent Reviewing:**

3

---

> ### Author Response · Authors · 2021-08-11
> **Response to Reviewer 1ACw**
>
> We would like to thank the reviewer for the careful review and thoughtful suggestions. We discuss all remaining questions and concerns below.
>
> **Experimental Evaluation**
> We challenge the view that there is not enough empirical evidence to justify the significance of the paper: our synthetic data is directly sampled from common stochastic processes, the experiments are carefully designed, and our results demonstrate improvements of the proposed model over multiple state-of-the-art methods; see Section 5.1. For example, geometric Brownian motion is a process that can be obtained by applying a non-lipschitz (exponential) transformation to a Wiener process, the linear SDE is not in the class of SDEs specified by Equation (3) (i.e., cannot be represented by CTFP transformations), and the continuous auto-regressive (CAR) model is a non-Markov continuous stochastic process.
>
> We believe an evaluation of negative log-likelihood on this controlled data is the purest way of demonstrating our model’s improvements over the baselines. Experimental results on real-world datasets, while important, also include factors beyond modelling of continuous dynamics based on partial observations; please refer to [VRNN Performance] in our response to Reviewer 73E9 for a more detailed discussion.
>
> VRNN is included for transparency and completeness but, while performing well on isolated test points, is unable to model continuous dynamics, which is the core contribution of CLPF. Among all models capable of learning continuous dynamics our model shows competitive (and in many cases superior) performance.
>
> We agree that a comparison of RMSEs could reaffirm the strengths of CLPF over these baselines. We will include such results on interpolation and prediction tasks in a separate post.
>
> **Dependence between $X_{t_i}$ and $Z_{t_{i-1}}$** [“Where does the dependency on Z_{t-1} come from?”]
> Regarding the dependence between $X_{t_i}$ and $Z_{t_{i-1}}$, please note that $X_{t_i}$ depends on both $O_{t_i}$ and $Z_{t_i}$. The distribution of $O_{t_i}$ depends on the value of $O_{t_{i-1}}$, which is determined by $X_{t_{i-1}}$ and $Z_{t_{i-1}}$. Omitting the $O_{t_i}$’s from the graphical model thus results in dependencies of $X_{t_i}$ on both $Z_{t_{i-1}}$ and $X_{t_{i-1}}$. We will add a detailed explanation in the revision of the paper and also clarify these dependencies in the graphical models (Figure 2).
>
> **Readability: Contributions and Algorithm in Pseudocode**
> We have a paragraph on the contributions of our work in Section 1 and will improve the clarity of this paragraph in the revision of the work. We would also like to thank the reviewer for the advice to include a description of CLPF in pseudocode to improve the readability of the paper.
>
> **Limitations and Social Impact**
> Limitations: While being more expressive than the CTFP model, the proposed CLPF model is not capable of exact likelihood evaluation and relies on variational approximations for estimation and optimization. The CLPF model can also comprise up to two differential equations that need to be numerically solved: the stochastic differential equation ($Z_t$) and the augmented neural ODE in the continuously-indexed normalizing flow mapping ($O_t$ and $X_t$), incurring higher computational cost. Fortunately, this computational cost can be controlled via additional parameters: (1) The approximation error of numerical solutions to SDEs/ODEs, as well as the inversion of a residual flow, can be controlled through the tolerance parameters of numerical solver and the number of fixed point iterations [1]; (2) The tightness of the IWAE bound for likelihood estimation can be controlled through the number of latent samples.
>
> Social impact: As an expressive generative model for continuous time series, we believe CLPF can be used to model continuous dynamics from partial observation in a wide range of areas, including physics, healthcare, and finances. Furthermore, it can potentially be leveraged in a variety of downstream tasks, including data generation, forecasting, and interpolation. However, we are also concerned that abuse of CLPF models for inferring unobserved time points could lead to undesired inference of restricted data. Protection of critical unobserved information is an active area of research [2] that can be explored as an orthogonal component in future versions of our model.
>
> [1] Behrmann, Jens, et al. "Invertible residual networks." International Conference on Machine Learning. PMLR, 2019.
>
> [2] M. Abadi et al. “Deep Learning with Differential Privacy”. Proceedings of the 2016 ACM SIGSAC conference on computer and communications security, 2016.

---

> > ### Comment · Reviewer_1ACw · 2021-08-16
> > **I keep my score**
> >
> > Thanks for your response, which satisfied me about the limitation, societal impact, and the X-Z dependency issues.
> >
> > However, the main issue in my review is that the proposed method does not demonstrate numerical performance superior to the state of art, remains untouched. The proposed method is behind or only on par with existing methods. Actually, it should have been possible to report the RMSEs of your already existing experiments during the rebuttal with negligible effort.
> >
> > Under these conditions, I do not see a reason to change my score.

---

> > ### Author Response · Authors · 2021-08-25
> > **Response Updates to Reviewer 1ACw**
> >
> > We would like to thank the reviewer for the timely feedback and provide a few updates to further address the reviewer’s concerns.
> >
> > **Experimental Evaluation**
> >
> > While analyzing the differences in real-world performance between VRNN and CLPF we discovered inconsistency in our experimental protocols that gives an advantage to VRNN. Aligning the protocols results in significant performance improvements of our model and almost completely eliminates the gap to VRNN. Code to reproduce the experiment and its results will be published together with the paper.
> >
> > We also evaluated the models using RMSE as the metric on sequential prediction tasks. Observations from the real-world datasets are fed as input to the model sequentially and we use the model to predict the immediate next observations based on the average of 125 samples. For all the latent variable models including VRNN, latent ODE, latent SDE, latent CTFP,  and CLPFs, the latent states for prediction are sampled from the variational posterior distribution conditioned on the observations before the prediction time stamp. The test sequences are generated in the same way as in the negative log-likelihood evaluation. Please refer to Sec. 3 of the supplementary material for more details of test sequence generation.
> >
> > The experiment results after protocol realignments are shown as follows: In Table 1, we show quantitative evaluation results on real-world datasets. In Table 2, we show the RMSE results on real-world datasets.
> >
> > Table 1 Real-World Quantitative Evaluation (Negative log-likelihood)
> >
> > | Model             | Mujoco[5] | BAQD[6] | PTBDB[7] |
> > |-------------------|-----------|---------|----------|
> > | VRNN[4]           | -15.122   | -1.192  | -2.012   |
> > | Latent ODE[2]     | 23.632    | 2.632   | -1.012   |
> > | Latent SDE[3]     | 3.210     | 1.224   | -1.158   |
> > | CTFP[1]           | -6.318    | -0.093  | -1.281   |
> > | Latent CTFP[1]    | -12.097   | -0.487  | -1.692   |
> > | CLPF-ANODE (Ours) | -14.013   | -0.693  | -1.521   |
> > | CLPF-iRes (Ours)  | -3.875    | -0.462  | -1.434   |
> >
> > Table 2 Real-World Quantitative Evaluation (RMSE). We report the mean with 25% and 75% quantiles in the square brackets in each cell.
> >
> > | Model             | Mujoco[2]               | BAQD[5]               | PTBDB[6]              |
> > |-------------------|-------------------------|-----------------------|-----------------------|
> > | VRNN[4]           | 1.683, [0.237, 1.089]   | 0.519, [0.166, 0.694] | 0.036, [0.005, 0.033] |
> > | Latnet ODE[2]     | 14.039, [9.915, 15.760] | 1.448, [0.973, 1.771] | 0.331, [0.220, 0.456] |
> > | Latent SDE[3]     | 7.059, [2.391, 7.868]   | 0.849, [0.445, 1.501] | 0.095, [0.029, 0.117] |
> > | CTFP[1]           | 2.082, [0.188, 1.888]   | 0.633, [0.142, 0.886] | 0.057, [0.005, 0.046] |
> > | Latent CTFP[1]    | 2.016, [0.167, 1.837]   | 0.675, [0.189, 0.938] | 0.069, [0.007, 0.065] |
> > | CLPF-ANODE (Ours) | 1.575, [0.154, 1.486]   | 0.548, [0.152, 0.755] | 0.048, [0.005, 0.042] |
> > | CLPF-iRes (Ours)  | 2.697, [0.523, 3.245]   | 0.574, [0.177, 0.789] | 0.055, [0.006, 0.048] |
> >
> > We hope the updated evaluation results with the original response can help address the reviewer’s concern on experiment results.
> >
> > [1] Deng, Ruizhi, et al. "Modeling continuous stochastic processes with dynamic normalizing flows." 34th Conference on Neural Information Processing Systems
> >
> > [2] Rubanova, Yulia, et al. "Latent Ordinary Differential Equations for Irregularly-Sampled Time Series." 33th Conference on Neural Information Processing Systems
> >
> > [3] Li, Xuechen, et al. "Scalable gradients for stochastic differential equations." International Conference on Artificial Intelligence and Statistics. PMLR, 2020.
> >
> > [4] Chung, Junyoung, et al. "A recurrent latent variable model for sequential data." 29th Conference on Neural Information Processing Systems
> >
> > [5] Zhang, Shuyi, et al. "Cautionary tales on air-quality improvement in Beijing." Proceedings of the Royal Society A: Mathematical, Physical and Engineering Sciences 473.2205 (2017): 20170457.
> >
> > [6] Bousseljot, R., D. Kreiseler, and A. Schnabel. "Nutzung der EKG-Signaldatenbank CARDIODAT der PTB über das Internet." (1995): 317-318.

---

### Official Review · Reviewer_73E9 · 2021-07-13

**Rating:** 8
**Confidence:** 3

**Summary:**

This paper introduces CLPF---novel parametrization of a latent SDE---that has higher expressivity than previous approaches. CLPF is obtained as a time-dependent invertible transformation of a base process.
This transformation is implemented as a normalizing flow whose parameters are a function of another SDF.
In addition, the authors introduce a novel non-Markov piecewise variational posterior process that elegantly circumvents the fact that SDE solutions are Markov. The method is evaluated on four well-designed syntethic datasets, whose purpose is to test different properties of CLPF. The efficacy to model more complicated data is tested on a real-world dataset, while an ablation study evaluates the relative importance of different model components.

**Limitations And Societal Impact:**

Authors do not discuss any limitations of this work (e.g. computational or memory cost, where it would fail compared to alternatives). A societal-impact discussion is absent but also not necessary for this paper.

**Main Review:**

Originality:
- This work combines several existing approaches and algorithmic improvements. Namely, CLPF extends Continuous-Time Flow Process (CTFP). CTFP is defined as an invertible transformation of a base process. This work changes the base process from Wiener to OU, which prevents the variance of the process from growing over time. It also introduces a time-varying flow, whose parameters are a function of another (latent) SDE. Finally, the authors introduce a novel piecewise variational posterior process which improved learning of the generative model and inference.
- It is clear how this work differs from the predecessors, which are adequately cited (I might have missed some relevant works as I'm not an expert in neural SDEs).

Quality:

- I am not familiar with SDE-based ELBOs, so I cannot comment on the correctness of the derivations.

The good:
- This paper seems to be well done.
- It is well written. The method is clearly explained.
- The experiments are well designed---I especially like the synthetic data experiments, where each experiment has a clear purpose (as it tests a different aspect of the model).
- The piecewise posterior approximation is novel and it is cleverly designed to circumvent a known limitation.

The bad:
- I think that in Eq 15, when going from line 1 to line 2, the authors missed the expectations with respect to previous latent variables. This is a common thing to not include them in sequential VAEs. However, this is incorrect if not stated explicitly.
- The authors do not discuss any limitations of the method. I wonder what they are, if any?
- In experiments, the VRNN (a discrete-time baseline) often outperforms continuous methods in terms of the marginal likelihood. I find that weird. While the authors mention that VRNN is unable to produce samples consistent with the likelihood, I do not understand why that is, and it is not explained.
- Fig. 3 contains qualitative evaluation but compares the proposed method only with a weak discrete-time baseline. Please include a better continuous-time alternative.

Questions:
- NODEs are notoriously slow and expensive to compute. CLPFs include three different neural SDEs (base for the likelihood, posterior and a latent one). How does that affect computational cost with respect to a vanilla NODE? How about with respect to VRNN (used as a baseline)?
- In a discrete-time system, a state could be engineered such that a Markov system becomes a non-Markov due to inclusion of additional information in the state. While I understand that your piecewise posterior idea solves the issue, I wonder if a similar solution to a discrete-time process could also be employed?
- Have you thought about using a deterministic neural ODE as an encoder instead of a discrete-time RNN? I think this could gracefully handle non-uniform temporal sampling. This could be formulated as integrating some hidden state between observation time-stamps, and discontinuous state updates (conditioned on an observation) at time-stamps with available observations.

Clarity:

- The paper is generally well-written and contains a sufficient amount of information about the method. It does not contain any details about the neural nets used or optimisation. The authors mention that the details are deferred to the appendix, but I haven't checked.
- It would be nice to have a few comments on the interpretation of certain quantities, e.g., what does M_T in Eq. 2 correspond to? Can I interpret M^{(i)} in Eq. 14 as an importance weight? Was does the u_2 - u_1 term in L82 mean?
- How do I interpret Eq. 3? As a non-expert in ODEs, I do not understand why that form is limiting.
- Eq 11 and 13 could be easier to read with a change of variables e.g. \hat{t_i} = t_{i_1} + \tilde{t} or similar.

Significance:
- The problems studied in this paper are currently very popular within the community.
- This work is quite original and significantly improves on prior art by improving benchmark results as well as meaningfully expanding the methodological toolkit.
- In my opinion this is a very significant piece of work.



**Time Spent Reviewing:**

6

---

> ### Author Response · Authors · 2021-08-11
> **Response to Reviewer 73E9**
>
> We would like to thank the reviewer for carefully reviewing our work, recognizing its originality and quality, and providing constructive comments. Below we respond to the remaining questions and concerns:
>
> **Expectation with Respect to Latent Variables** [“I think that in Eq 15, when going from line 1 to line 2, the authors missed the expectations with respect to previous latent variables.”] We acknowledge that there is a lack of clear description on the relations between $\tilde{z}\_{t_i}$s and $\omega_i$s. In Equation (15), the exact value of the latent variable $\tilde{z}\_{t_i}$ can be determined by the value of $\tilde{z}\_{t_{i-1}}$ and the sample path of the Wiener processes from $t_{i-1}$ to $t_i$, $\omega_i$. Therefore, we can sample $\tilde{z}\_{t_i}$ given the initial value $\tilde{z}\_{t_0}$ and sample paths of Wiener processes up to $t_i$, i.e. $\omega_1, \omega_2, \dots, \omega_i$. We will clarify this relationship in the revision of the paper.
>
> **VRNN Performance** [“the VRNN (a discrete-time baseline) often outperforms continuous methods in terms of the marginal likelihood.”] The performance of statistical models is affected by a variety of factors. In our problem setting for real-world datasets, we assume that the observed values are updated at a fixed rate as mentioned in Section 5.2. Therefore, we are approximating continuous dynamics with a discrete real-world dataset, which might favour discrete models like VRNN over continuous-time models, especially considering that VRNN takes  the observation time stamp $t_i$ and the difference between timestamps $t_i - t_{i-1}$ as inputs. However, VRNN models do not define a continuous-time stochastic process and the consistency of observational distributions on discrete time grids are not guaranteed. For example, VRNN can induce a distribution $p_{t_1, \dots, t_i-1, t_i, t_{i+1},\dots, t_n}(\cdot)$ on the time grid ${t_1, \dots, t_{i-1}, t_i, t_{i+1},\dots, t_n}$and another distribution $p_{t_1, \dots, t_i-1, t_{i+1},\dots, t_n}(\cdot)$ on the time grid ${t_1, \dots, t_{i-1}, t_{i+1},\dots, t_n}$. However, it is not guaranteed that $\int p_{t_1, \dots, t_i-1, t_i, t_{i+1},\dots, t_n}(X_{t_1}, \dots, X_{t_{i-1}}, X_{t_i}, X_{t_{i+1}},\dots, X_{t_n}) \textrm{d} x_{t_i}$ equals $p_{t_1, \dots, t_i-1, t_{i+1},\dots, t_n}(X_{t_1}, \dots, X_{t_{i-1}}, X_{t_{i+1}},\dots, X_{t_n})$. In practice, this corresponds to sampling $X_{t_1}, \dots, X_{t_{i-1}}, X_{t_i}, X_{t_{i+1}},\dots, X_{t_n}$ and dropping $X_{t_i}$ versus directly sampling $X_{t_1}, \dots, X_{t_{i-1}}, X{t_{i+1}},\dots, X_{t_n}$; samples from these two approaches are not guaranteed to follow the same distribution. In contrast to that, CLPF and the other baseline models define continuous-time stochastic processes, i.e., consistency is guaranteed for them. Please refer to the marginalization consistency condition of the Kolmogorov extension theorem [1, Theorem 2.1.5] for a more general statement. The primary purpose of showing qualitative samples of VRNNs and comparing it with our model is to highlight the difference between the two models in defining continuous dynamics and the consistency of distributions. To generate plots of continuous trajectories, we use a dense time grid to approximate a continuous time interval while the models are trained with observations on much sparser grids. We will present qualitative descriptions of samples from the other models in a later response and include the corresponding figures in the revision of the paper.
>
> **Computational Cost and Evaluation Time**
> The encoders of latent ODE and latent CTFP models use an RNN-ODE [2], which comprises an ODE and an RNN module to encode irregular time series data. In contrast, our implementations of CLPF and latent SDE employ a vanilla RNN to encode irregular time series observations, which is faster than the RNN-ODE. Furthermore, although we apply the same regularization during training, the dynamics learned by the augmented neural ODE could be different for each model, which also affects the runtime. For density estimation, we need to solve at most two differential equations to compute the term in Equation (4): first we solve an augmented version of the posterior SDE that computes the importance term and samples the latent trajectory, then we solve the augmented neural ODE of the continuously-indexed normalizing flow. We will include an overview of runtimes for each model in a separate post.
>
> **Interpretation of $M_T$** [“Can I interpret M^{(i)} in Eq. 14 as an importance weight?”] The terms $M^{(i)}$ and $M_T$ can be interpreted as the importance terms of trajectories sampled from two different continuous-time stochastic processes. In line 82, $\mu_1$ and $\mu_2$ are the drift functions of the prior process and the posterior process defined by the SDEs in line 79 and line 80, respectively, and $\sigma$ is their common variance function.
>
> **Discrete Non-Markov Processes and Neural ODE as Encoder** We agree with the reviewer on the suggestion to construct a posterior for a discrete process and to use a neural ODE as a data encoder and believe they are worth exploring in a future work.
>
> **Implications of Equation (3)** Please find Theorem 1 in the supplementary material for an example of a stochastic process whose corresponding SDE cannot be written in the form of Equation (3).
>
> [Equation (11) / Equation (13) Readability] We also want to thank the reviewer for the suggestion on the presentation of Equation (11) / Equation (13) and will incorporate these changes in the revision of the paper.
>
> **Limitations and Social Impact** We will add the following discussion on our model’s limitations and social impact:
>
> Limitations: While being more expressive than the CTFP model, the proposed CLPF model is not capable of exact likelihood evaluation and relies on variational approximations for estimation and optimization. The CLPF model can also comprise up to two differential equations that need to be numerically solved: the stochastic differential equation ($Z_t$) and the augmented neural ODE in the continuously-indexed normalizing flow mapping ($O_t$ and $X_t$), incurring higher computational cost. Fortunately, this computational cost can be controlled via additional parameters: (1) The approximation error of numerical solutions to SDEs/ODEs, as well as the inversion of a residual flow, can be controlled through the tolerance parameters of numerical solver and the number of fixed point iterations [3]; (2) The tightness of the IWAE bound for likelihood estimation can be controlled through the number of latent samples.
>
> Social impact: As an expressive generative model for continuous time series, we believe CLPF can be used to model continuous dynamics from partial observation in a wide range of areas, including physics, healthcare, and finances. Furthermore, it can potentially be leveraged in a variety of downstream tasks, including data generation, forecasting, and interpolation. However, we are also concerned that abuse of CLPF models for inferring unobserved time points could lead to undesired inference of restricted data. Protection of critical unobserved information is an active area of research [4] that can be explored as an orthogonal component in future versions of our model.
>
> [1] B. Oksendal. “Stochastic Differential Equations: An Introduction with Applications”. Springer Science & Business Media, 2013.
>
> [2] Y. Rubanova, RTQ Chen, and D. Duvenaud. “Latent ODEs for Irregularly-Sampled Time Series.” NeurIPS, 2019.
>
> [3] Behrmann, Jens, et al. "Invertible residual networks." International Conference on Machine Learning. PMLR, 2019.
>
> [4] M. Abadi et al. “Deep Learning with Differential Privacy”. Proceedings of the 2016 ACM SIGSAC conference on computer and communications security, 2016.

---

> > ### Comment · Reviewer_73E9 · 2021-08-23
> > **Keeping my score.**
> >
> > Thanks for the response, it does answer my questions.

---

> > ### Author Response · Authors · 2021-08-25
> > **Response Updates to Reviewer 73E9**
> >
> > We would like to thank the reviewer for the encouraging feedback and provide a few updates on our response to the reviewer’s concerns.
> >
> > **Computational Cost and Evaluation Time**
> > We calculated the average of wall-clock evaluation times of 5 batches in the Mujoco dataset for each model. The batch size is 5 and the latent sample size is 125 which is the same as our setting for evaluation. Please refer to Section 3 in the supplementary material for more details. The evaluation times in seconds are reported in the Table 1 below.
> >
> > Table 1 Evaluation Times of Models
> >
> > | VRNN[4]     | Latent ODE[2] | Latent SDE[3] | CTFP[1]     | Latent CTFP[1] | CLPF-ANODE (Ours) | CLPF-iRes (Ours) |
> > |-------------|---------------|---------------|-------------|----------------|-------------------|------------------|
> > | 2.042+0.393 | 0.269+0.051   | 1.608+0.120   | 7.620+0.085 | 10.199+0.895   | 19.730+1.333      | 12.596+0.809     |
> >
> > **Qualitative Samples of Other Baseline Models**
> > We generated sample trajectories for the other baseline models trained on BAQD datasets and plotted the second dimension of the samples. We also discovered inconsistencies in our real-world data experimental protocols which give disadvantages to our CLPF models. We regenerated the sample trajectories for VRNN and CLPF models after realigning the protocols. Please refer to **Experimental Evaluation** in **Response Updates to Reviewer 1ACw** for the updated experiment results on real-world datasets. Qualitative descriptions of the sample trajectories are presented below:
> >
> > * CTFP: We find CTFP is able to generate trajectories continuous in time. The variance of the trajectories’ values grows as time moves on and is larger than samples from the other models and ground truth data. This change of variance in the sampled trajectories reflects the change of variance in the base Wiener process. We also find there’s a lack of change of trends within each trajectory sample compared with samples from ground truth data, CLPF and, Latent SDE.
> >
> > * Latent CTFP: Latent CTFP is also able to generate trajectories continuous in time. There’s a less significant growth of variance in the sampled trajectories compared to CTFP model. But similar to the CTFP model, the latent CTFP model also shows a lack of trajectory trend variations compared with samples from ground truth data, CLPF, and Latent SDE.
> >
> > * Latent ODE: Latent ODE is unable to generate continuous trajectories. The trajectories generated by latent ODE show very large changes between two consecutive steps. This is potentially due to predictions of large variance when decoding the latent trajectories to observational distributions. The change of values between steps often suppresses the trends of the trajectories and variance of values across trajectories.
> >
> > * Latent SDE: Latent SDE is unable to generate continuous trajectories either but the variance of values between two consecutive steps is much smaller than latent ODE. Latent SDE also shows more significant changes of trends within each trajectory than CTFP and latent CTFP models but less significant than the ground truth data and the CLPF model.
> >
> > * VRNN: As VRNN does not define distributions consistently on different time grids, we generate samples from VRNN on 3 different time grids in the interval between 0 and 30. All the time grids are regular and the time gaps between consecutive steps are set to 0.01, 0.2, and 0.5. However, the generated trajectories show different visual patterns with different time gaps.
> >
> >     * When the time gap is 0.01, the regenerated samples from VRNN model show both similarity and difference to the VRNN samples in the original submission which are also generated with time gap equaling to 0.01. We do observe that some regenerated trajectories show variations of trends that are similar to CLPF and ground truth samples. However, most samples stop showing changes of trends and each of them only vibrates around a certain value with small variations which is also observed in Figure 2(b) in the submission.
> >
> >     * When time gap is 0.2, the generated samples from VRNN show diverse changes of trends similar to latent SDE but the generated trajectories look smoother than the samples from latent SDE. The variance of values across different trajectories is similar to the ground truth data.
> >
> >     * The trajectories generated with a time gap of 0.5 between consecutive steps also show a lack of trend change; the trajectories look relatively flat compared to other models. Values across trajectories show a level of variance similar to the samples generated with time gap of 0.2 and the ground truth samples.
> >
> > * CLPF: The regenerated samples from the CLPF model have similar visual appearance as the samples presented in Figure 2(a) of the original submission. The sample trajectories are continuous in time and they also show similar patterns of trend changes to the ground truth data.
> >
> > [1] Deng, Ruizhi, et al. "Modeling continuous stochastic processes with dynamic normalizing flows." 34th Conference on Neural Information Processing Systems
> >
> > [2] Rubanova, Yulia, et al. "Latent Ordinary Differential Equations for Irregularly-Sampled Time Series." 33th Conference on Neural Information Processing Systems
> >
> > [3] Li, Xuechen, et al. "Scalable gradients for stochastic differential equations." International Conference on Artificial Intelligence and Statistics. PMLR, 2020.
> >
> > [4] Chung, Junyoung, et al. "A recurrent latent variable model for sequential data." 29th Conference on Neural Information Processing Systems

---

### Official Review · Reviewer_XbGb · 2021-07-14

**Rating:** 7
**Confidence:** 4

**Summary:**

This paper proposes a continuous latent time series model called the Continuous Latent Process Flow (CLPF). Irregular time series observations are modeled as invertible transformations (i.e. normalizing flows) of an underlying stochastic process (the base process), where the flow transformation is itself parameterized by a time-varying stochastic process (the latent process), modeled by a (learned) stochastic differential equation (SDE). The authors also develop a piecewise variational approximation to the latent process, which they train by maximizing the importance-weighted ELBO. In contrast to prior work like Continuous Time Flow Processes (CTFP) and latent SDES, the authors show that CLPFs better model synthetic and real-world time series, including processes which require non-Lipschitz transforms of base processes, and non-Markovian processes. CLPFs also better capture qualitative trends in real world data.

**Limitations And Societal Impact:**

I think this paper could do with more discussion of limitations, especially relative to existing methods. Besides the issues with numerical approximation that were (very briefly) noted, one potential limitation that comes to mind is that CLPFs have a higher training, likelihood evaluation, and inference cost, due to the absence of a tractable likelihood. This is in contrast to CTFPs, which can take advantage of a deterministic transformation between the base and observed process, and do not require variational inference for training. Presumably this means that CLPFs might require more time / compute / energy train to reach the same performance on some tasks, and to perform likelihood evaluations, which might motivate using other models (even if they are less flexible).

The authors did not discuss societal impacts. I would encourage them to do so -- perhaps there are positive/negative impacts of being able to model real-world continuous time data more faithfully, e.g. for forecasting and scientific modeling -- but I think it is their prerogative, given that this is more of a "general methodology" paper rather than an application-focused paper.

**Main Review:**

This was a well written paper that by-and-large demonstrated its quality beyond existing methods, both empirically and mathematically. While the clarity of the figures and exposition of the math could use a bit of improvement, the experiments are thorough, and are well-designed to illustrate the differences between CLPFs and prior work, making the work of significance to the field. I especially appreciated the inclusion of ablation studies to show the importance of each component. The use of an SDE to parameterize a flow-based transform is also original, to my knowledge, as is the piece-wise variational approximation. In light of the originality, quality and significance of the proposed method, I recommend acceptance of the paper.  Greater clarity would have improved my score to an 8. I just have some clarificatory questions, and some suggestions to improve how the model and its math are communicated.

My main clarificatory question is why it's important to have _both_ a base process (which is transformed to the observed process) and a latent process (which parameterizes the transformation) in order to achieve modeling flexibility. I understand how this choice improves over CTFPs, which do not handle non-Lipschitz transformations explicitly. But what about something like the latent SDE with suitable augmentations:
1. Model the base process using an SDE (instead of the OU or Wiener process)
2. Transform that using a CTFP-style continuously-indexed flow to the observable process
3. Use a piecewise SDE variational approximation so that the posterior can be non-Markovian

Would the worry about non-Lipschitz transforms apply to this? (And is this actually one of the ablated baselines tested? I couldn't tell). Because this would avoid the need to have two unobserved stochastic processes, which seems like it would reduce computational cost. Basically, I think it'd be great to better explain why you need both the $O_t$ process and the $Z_t$ process, instead of just transforming the $Z_t$ process to the $X_t$ process directly. (The other alternative is transforming the $O_t$ process to the $X_t$ process directly, which I understand would be a variant of CTFP.)

Apart from that, I wanted to share a few points that were confusing to me about the figures and the math. For Figure 1, it was not immediately obvious to me, without reading the paper closely, that the latent process wasn't the same as the base process. It's also generally not clear to me what is illustrated on the right-hand-side of the picture. Are the grey lines showing (stochastic) transformations) of specific values of the base process to the observed process? Is the blue line showing the evolution of the observed process, or something else? I think it'd be helpful to show samples of both the base process and the observed process (like in the CTFP) paper, and then on top of that show how the latent process parameterizes transformations from one sample of the base process to one sample of the observed process.

For Figure 2, the graphical model was confusing to me, because it omits / marginalizes out the base process $O_t$. Intuitively, I was thinking of a CLPFs as a HMM-style graphical model with *two* hidden states at each $t$, $O_t$ and $Z_t$, and so I was surprised to see the non-Markovian dependence of $X_{t_i}$ on $Z_{t_{i-1}}$. It was only after I thought a while about the math shown on Eq. 9 that I realized that marginalizing out $O_t$ leads to the dependence of $X_{t_i}$ on $Z_{t_{i-1}}$, but this wasn't clearly shown / derived. I think it would be a good idea to revise both the Equations and Figure 2 to better explain how this dependence arises.

Finally, I think it might be a good idea to call the reweighting term $M^{(i)}$ in Eq. 14 an importance weight, since that is a readily recognized term for reweighting samples from a proposal distribution (in this case, the approximate posterior).

== Post-Rebuttal ==

Thank you to the authors for their detailed responses, clarifications, and additional experiments. I continue to believe this is significant paper that makes meaningful methodological contributions to the field, and I think this has been further demonstrated by the additional experiments. I also appreciate that the author's are aware of the method's limitations and social impact, and look forward to seeing their inclusion in the final paper. I will maintain my score of 7.

**Time Spent Reviewing:**

6

---

> ### Author Response · Authors · 2021-08-11
> **Response to Reviewer XbGb**
>
> We would like to thank the reviewer for recognizing the quality and contributions of our work and providing thoughtful comments in the review. Questions and feedback are addressed below.
>
> **Importance of $Z_t$ and $O_t$** [“why it's important to have both a base process (which is transformed to the observed process) and a latent process”] We want to clarify the importance of having both a latent process $Z_t$ and a base process $O_t$ from the following perspectives:
>
> 1. As many latent variable models, including hidden Markov models and variational auto-encoders [1], the two stochastic processes $Z_t$ and $O_t$ can be viewed as two sources of stochasticity: the latent state and the observational noise. As a common practice in VAEs, a sample of the latent state is decoded into a more expressive observational distribution through reparameterization of a standard Gaussian distribution by proposing the mean (shift) and standard deviation (scale). In our model, a sample trajectory of the stochastic process is decoded into a more expressive observational process by deforming the base process $O_t$ using indexed normalizing flows **driven** by $Z_t$. If we directly applied CTFP-style continuously-indexed normalizing flows to a base process modeled by an SDE, the observed process defined by this generative model would be constrained to a Markov process.
>
> 2. Decoding samples of $Z_t$ by transforming $O_t$ using invertible mappings leads to closed-form observational distributions conditioned on sample trajectories of $Z_t$. For most continuous-time stochastic processes modeled by an SDE, closed-form transition distributions between two timestamps $p(z_{t_i} | z_{t_{i-1}})$ do not exist. OU processes and Wiener processes are special cases with simple closed-form transition distributions $p(O_{t_i} | O_{t_{i-1}})$ that can be used to derive the likelihood of observations conditioned on samples of $Z_t$ and $p(x_{t_1}, x_{t_2}, \dots, x_{t_n}|z_t(\omega))$, as shown in Equation (9) and Equation (10). Therefore, we can rewrite the observational likelihood $p(x_{t_1}, x_{t_2}, \dots, x_{t_n})$ as an expectation of $p(x_{t_1}, x_{t_2}, \dots, x_{t_n} | z_t(\omega))$ and train the model in a variational Bayes framework. The variational Bayes framework or maximum likelihood training may not be directly applicable to the suggested method due to the following difficulties: (a) As the base and variational posterior processes are both defined by SDEs, the transition densities of the base process and observational process, $p(z_{t_i} | z_{t_{i-1}})$ and $p(x_{t_i} | x_{t_{i-1}})$, do not have closed forms; and (b) CTFPs are one-to-one mappings, i.e., the distribution $p(x_{t_i} | z_{t_i})$ is a Dirac-delta distribution and poses challenges for optimization in this framework unless some observational noise is introduced for $X_{t_i}$, which can be equivalently viewed as applying another decoder on top of $X_t$.
>
> 3. Trajectories of $Z_t$ and $O_t$ are guaranteed to be continuous and transforming them with CTFP-style continuously-indexed normalizing flows guarantees that we can sample continuous trajectories from our model. We believe this is a better inductive bias for the model when fitting data sampled from partial observations of continuous dynamics than modeling static distributions at each individual time stamp.
>
> 4. Finally, we would like to point out that directly applying CTFP-style transformations to a stochastic process defined by an SDE would result in another SDE, which can be shown using Ito’s Lemma [2, Theorem 4.2.1].
>
> We will include these discussions in the revision of the paper.
>
> **Figures and Terminology** We would like to thank the reviewer for the suggestions on the presentation of the paper. The grey lines show the individual trajectories of the transformation from the base process to the observed process, driven by the augmented neural ODE. The blue lines on the right-hand side show the evolution of the observed process. Out of clarity, the blue line on the left-hand side shows a sample latent process. Improvements to Figure 1 will be incorporated in the revision of the paper. A more detailed explanation of the dependencies between $z_{t_{i-1}}$ and $z_{t_i}$, including the role of $O_t$ in the graphical models (Figure 2), will also be added. We agree that “importance weight” is a more fitting term for $M(i)$.
>
> **Limitations and Social Impact** We will add the following discussion on our model’s limitations and social impact:
>
> Limitations: While being more expressive than the CTFP model, the proposed CLPF model is not capable of exact likelihood evaluation and relies on variational approximations for estimation and optimization. The CLPF model can also comprise up to two differential equations that need to be numerically solved: the stochastic differential equation ($Z_t$) and the augmented neural ODE in the continuously-indexed normalizing flow mapping ($O_t$ and $X_t$), incurring higher computational cost. Fortunately, this computational cost can be controlled via additional parameters: (1) The approximation error of numerical solutions to SDEs/ODEs, as well as the inversion of a residual flow, can be controlled through the tolerance parameters of numerical solver and the number of fixed point iterations [3]; (2) The tightness of the IWAE bound for likelihood estimation can be controlled through the number of latent samples.
>
> Social impact: As an expressive generative model for continuous time series, we believe CLPF can be used to model continuous dynamics from partial observation in a wide range of areas, including physics, healthcare, and finances. Furthermore, it can potentially be leveraged in a variety of downstream tasks, including data generation, forecasting, and interpolation. However, we are also concerned that abuse of CLPF models for inferring unobserved time points could lead to undesired inference of restricted data. Protection of critical unobserved information is an active area of research [4] that can be explored as an orthogonal component in future versions of our model.
>
> [1] DP Kingma and M. Welling. “Auto-Encoding Variational Bayes”. ICLR, 2014.
>
> [2] B. Oksendal. “Stochastic Differential Equations: An Introduction with Applications”. Springer Science & Business Media, 2013.
>
> [3] Behrmann, Jens, et al. "Invertible residual networks." International Conference on Machine Learning. PMLR, 2019.
>
> [4] M. Abadi et al. “Deep Learning with Differential Privacy”. Proceedings of the 2016 ACM SIGSAC conference on computer and communications security, 2016.

---

### Decision · Program_Chairs · 2021-09-27

**Decision:**

Accept (Poster)

**Comment:**

This paper explores a continuous latent time series model, the Continuous Latent Process Flow (CLPF).

CLPFs can be seen as a generalisation of CTFPs where, instead of transforming a base Wiener process with a time-conditional flow, it first introduces a latent process SDE which conditions a CLPF-like model (that uses an OU base process instead of a Wiener process). In that sense, CLPFs combine elements from VRNNs, latent SDEs and CTFPs.

In terms of motivation/discussions, it is clear why CLPFs should be strictly better than CTFPs, but it is not clear why they should be better than latent SDEs (at least not "a priori") so adding more discussions on this could improve the draft.

The experiments show good performance on real-world data relative to CTFPs, latent SDEs and VRNNs. In spite of some hesitance amongst reviewers, I find the experimental results compelling. But I also agree that the paper could benefit from more detailed experiments comparing to VRNNs, for instance to clarify if the difference with respect to VRNN is due to implementation differences (e.g. passing absolute time or not, or small architectural differences).